# Gradient Informed Proximal Policy Optimization

**Sanghyun Son**    **Laura Yu Zheng**    **Ryan Sullivan**    **Yi-Ling Qiao**    **Ming C. Lin**
Department of Computer Science
University of Maryland, College Park
{shh1295,lyzheng,rsulli,yilingq,lin}@umd.edu

## Abstract

We introduce a novel policy learning method that integrates analytical gradients from differentiable environments with the Proximal Policy Optimization (PPO) algorithm. To incorporate analytical gradients into the PPO framework, we introduce the concept of an $\alpha$-policy that stands as a locally superior policy. By adaptively modifying the $\alpha$ value, we can effectively manage the influence of analytical policy gradients during learning. To this end, we suggest metrics for assessing the variance and bias of analytical gradients, reducing dependence on these gradients when high variance or bias is detected. Our proposed approach outperforms baseline algorithms in various scenarios, such as function optimization, physics simulations, and traffic control environments. Our code can be found online: https://github.com/SonSang/gippo.

## 1 Introduction

Reinforcement learning (RL) techniques often take gradient-free experiences or trajectories to collect information and learn policies—suitable for many applications such as video games and recommender systems. In contrast, differentiable programming offers analytical gradients that describe the action-result relationship (Equation 2). In particular, recent developments in programming tools [Paszke et al., 2017] have enabled direct access to accurate analytical gradients of the environmental dynamics, as exemplified in physics simulations [de Avila Belbute-Peres et al., 2018, Degrave et al., 2019, Geilinger et al., 2020, Freeman et al., 2021, Qiao et al., 2021], instead of using differentiable models for approximation [Deisenroth and Rasmussen, 2011, Deisenroth et al., 2013]. Policy gradients estimated from the integration of these analytical gradients and the reparameterization trick (RP gradient) [Kingma and Welling, 2013] often display less variance than likelihood ratio policy gradients (LR gradient) [Williams and Peng, 1989, Glynn, 1990], derived using the log-derivative trick. Consequently, a subset of research has explored the utilization of these analytical gradients for RL applications, especially in physics domains [Qiao et al., 2021, Mora et al., 2021, Xu et al., 2022].

Nevertheless, in environments with inherent chaotic nature, the RP gradient is found to exhibit high variance [Parmas et al., 2018] and even empirical bias [Suh et al., 2022]. Therefore, a strategic approach is necessary for the adoption of analytical gradients in policy learning [Metz et al., 2021]. One such strategy involves the interpolation of the LR and RP gradients based on their variance and bias [Parmas et al., 2018, Suh et al., 2022], made possible by the fact that both are different estimators of the same policy gradient. However, this approach requires estimating the LR and RP gradient for each sample trajectory to compute sample variance, a process that can be time-consuming (Appendix 7.7). Additionally, for on-policy RL methods that do not initially estimate the LR gradient, such as Trust Region Policy Optimization (TRPO) [Schulman et al., 2015a] or Proximal Policy Optimization (PPO) [Schulman et al., 2017] algorithms, it remains uncertain how to evaluate the variance and bias of analytical gradients. These gradients, however, continue to be desirable when they exhibit low variance and provide insightful information about environmental dynamics.

37th Conference on Neural Information Processing Systems (NeurIPS 2023).

Given these challenges, our study explores how to incorporate analytical gradients into the PPO framework, taking their variance and bias into account. Our primary contributions are threefold: (1) we propose a novel method to incorporate analytical gradients into the PPO framework *without the necessity of estimating LR gradients*, (2) we introduce an adaptive $\alpha$-policy that allows us to dynamically adjust the influence of analytical gradients by evaluating the variance and bias of these gradients, and (3) we show that our proposed method, GI-PPO, strikes a balance between analytical gradient-based policy updates and PPO-based updates, yielding superior results compared to baseline algorithms in various environments.

We validate our approach across diverse environments, including classical numerical optimization problems, differentiable physics simulations, and traffic control environments. Among these, traffic environments typify complex real-world scenarios where analytical gradients are biased. We demonstrate that our method surpasses baseline algorithms, even under these challenging conditions.

## 2 Related Work

### 2.1 Policy Learning Without Analytical Gradients

The policy gradient is described as the gradient of the expected cumulative return in relation to policy parameters [Sutton et al., 1999]. For a stochastic policy, as examined in this paper, REINFORCE [Williams, 1992] represents one of the initial methods to employ a statistical estimator for the policy gradient to facilitate policy learning. The policy gradient obtained with this estimator is commonly referred to as a likelihood-ratio (LR) gradient [Glynn, 1990, Fu et al., 2015]. Despite the LR gradient's ability to operate without analytical gradients of dynamics and its unbiased nature, it frequently exhibits high variance [Sutton et al., 1998]. Furthermore, the nonstationarity of the policy renders stable learning difficult to guarantee [Konda and Tsitsiklis, 1999, Schulman et al., 2015b].

Several strategies have been proposed to address the high variance issue inherent to the LR gradient, including the use of additive control variates [Konda and Tsitsiklis, 1999, Greensmith et al., 2004]. Among these, the actor-critic method [Barto et al., 1983] is highly favored. However, the introduction of a value function may induce bias. To counteract this, the Generalized Advantage Estimator (GAE) [Schulman et al., 2015b] was developed to derive a more dependable advantage function. Concerning the issue of unstable learning, methods such as TRPO [Schulman et al., 2015a] and PPO [Schulman et al., 2017] propose optimizing a surrogate loss function that approximates a random policy's expected return. By constraining the policy updates to remain close to the current policy, these methods significantly enhance stability in learning compared to pure policy gradient methods, which makes them more preferred nowadays.

### 2.2 Policy Learning With Analytical Gradients

In contrast to the LR gradient, the RP gradient is based on the reparameterization trick, which mandates analytical gradients of dynamics yet typically achieves lower variance [Kingma and Welling, 2013, Murphy, 2022]. The RP gradient can be directly estimated over the entire time horizon using Backpropagation Through Time (BPTT) [Mozer, 1989] and subsequently used for policy updates. However, BPTT has been found to encounter multiple optimization challenges such as explosion or vanishing of analytical gradients over extended trajectories [Freeman et al., 2021]. This is because, surprisingly, the RP gradient, contrary to earlier beliefs, may experience exploding variance due to the chaotic nature of environments [Parmas et al., 2018, Metz et al., 2021]. Additionally, Suh et al. [2022] noted that the RP gradient could also be subject to empirical bias.

Several corrective measures exist for the issues concerning the RP gradient [Metz et al., 2021], such as using a truncated time window bootstrapped with a value function [Xu et al., 2022]. This approach has demonstrated its efficacy in addressing physics-related problems. Another class of solutions involves interpolating between LR and RP gradients by evaluating their sample variance and bias [Parmas et al., 2018, Suh et al., 2022]. Likewise, when we carefully address the problems of RP gradient, it can provide valuable information helpful for policy learning. However, when it comes to PPO, which is one of the most widely used RL methods nowadays, it is unclear how we can leverage the analytical gradients that comprise the RP gradient, as it does not estimate LR gradient which can be used for comparison to the RP gradient. In this paper, we address this issue and show that we can use analytical gradients to enhance the performance of PPO.

## 3 Preliminaries

### 3.1 Goal

We formulate our problems as a Markov decision process (MDP) defined by a tuple $(S, A, P, r, \rho_0, \gamma)$, where $S$ is a set of states, $A$ is a set of actions, $P : S \times A \times S \to \mathbb{R}$ is a state transition model, $r : S \times A \to \mathbb{R}$ is a reward model, $\rho_0$ is the probability distribution of the initial states, and $\gamma$ is a discount factor.

Under this MDP model, our goal is to train a parameterized stochastic policy $\pi_\theta : S \times A \to \mathbb{R}^+$, where $\theta$ represents parameters, that maximizes its expected sum of discounted rewards, $\eta(\pi_\theta)$:

$$\eta(\pi_\theta) = \mathbb{E}_{s_0, a_0, \ldots \sim \pi_\theta} \left[ \sum_{t=0}^{\infty} \gamma^t r(s_t, a_t) \right]. \tag{1}$$

In this paper, we denote the current policy that is used to collect experience as $\pi_{\bar{\theta}}$.

### 3.2 Analytical Gradients

We assume that $S$ and $A$ are continuous, differentiable spaces, and $P$ and $r$ are also differentiable models. Then, our differentiable environment provides us with following analytical gradients of observation or reward at a later timestep with respect to the action at the previous timestep:

$$\frac{\partial s_{t+1+k}}{\partial a_t}, \frac{\partial r_{t+k}}{\partial a_t}, \tag{2}$$

where $k \geq 0$ is an integer. With these basic analytical gradients, we can compute the gradient of certain advantage function, which we denote as $\frac{\partial A}{\partial a}$. In this paper, we use Generalized Advantage Estimator (GAE) [Schulman et al., 2015b]. See Appendix 7.2.1 for details.

### 3.3 Policy Update

In this section, we discuss how we update policy in two different settings: the policy gradient method based on RP gradient and PPO.

#### 3.3.1 RP Gradient

To compute the RP gradient, we rely on the reparameterization trick [Kingma and Welling, 2013]. This trick requires us that we can sample an action $a$ from $\pi_\theta$ by sampling a random variable $\epsilon$ from some other independent distribution $q$. To that end, we assume that we use a continuous differentiable bijective function $g_\theta(s, \cdot) : \mathbb{R}^n \to \mathbb{R}^n$ that maps $\epsilon$ to an action $a$, where $n$ is the action dimension. Then the following holds because of injectivity,

$$\left| \det(\frac{\partial g_\theta(s, \epsilon)}{\partial \epsilon}) \right| > 0, \forall \epsilon \in \mathbb{R}^n, \tag{3}$$

and we can define the following relationship.

**Definition 3.1** *We define $\pi_\theta \triangleq g_\theta$ if following holds for an arbitrary open set $T_\epsilon \subset \mathbb{R}^n$:*

$$\int_{T_a} \pi_\theta(s, a) da = \int_{T_\epsilon} q(\epsilon) d\epsilon,$$

*where $T_a = g_\theta(s, T_\epsilon)$.*

**Lemma 3.2** *If $\pi_\theta \triangleq g_\theta$,*

$$\pi_\theta(s, a) = q(\epsilon) \cdot \left| \det(\frac{\partial g_\theta(s, \epsilon)}{\partial \epsilon}) \right|^{-1},$$

*where $a = g_\theta(s, \epsilon)$. The inverse is also true.*

**Proof:** See Appendix 7.1.1.

Specifically, we use $q = \mathcal{N}(0, I)$ in this paper, and define $g_\theta$ as follows,

$$g_\theta(s, \epsilon) = \mu_\theta(s) + \sigma_\theta(s) \cdot \epsilon \quad (\|\sigma_\theta(s)\|_2 > 0),$$

which satisfies our assumption.

With $g_\theta(s, \epsilon)$ and the analytical gradients from Equation 2, we obtain RP gradient by directly differentiating the objective function in Equation 1, which we use for gradient ascent. See Appendix 7.2.2 for details.

### 3.3.2  PPO

PPO relies on the observation that we can evaluate a policy $\pi_\theta$ with our current policy $\pi_{\bar{\theta}}$, using its advantage function $A_{\pi_{\bar{\theta}}}$ as

$$\eta(\pi_\theta) = \eta(\pi_{\bar{\theta}}) + \int_s \rho_{\pi_\theta}(s) \int_a \pi_\theta(s, a) A_{\pi_{\bar{\theta}}}(s, a),$$

where $\rho_{\pi_\theta}(s)$ is the discounted visitation frequencies. See [Kakade and Langford, 2002, Schulman et al., 2015a] for the details.

As far as the difference between $\pi_{\bar{\theta}}$ and $\pi_\theta$ is sufficiently small, we can approximate $\rho_{\pi_\theta}(s)$ with $\rho_{\pi_{\bar{\theta}}}(s)$ [Schulman et al., 2015a], and get following surrogate loss function:

$$L_{\pi_{\bar{\theta}}}(\pi_\theta) = \eta(\pi_{\bar{\theta}}) + \int_s \rho_{\pi_{\bar{\theta}}}(s) \int_a \pi_\theta(s, a) A_{\pi_{\bar{\theta}}}(s, a). \tag{4}$$

Note that we can estimate this loss function with Monte Carlo sampling as described in Appendix 7.2.3. Since this is a local approximation, the difference between $\pi_{\bar{\theta}}$ and $\pi_\theta$ must be small enough to get an accurate estimate. TRPO [Schulman et al., 2015a] and PPO [Schulman et al., 2017] constrain the difference to be smaller than a certain threshold, and maximize the right term of (4). In particular, PPO restricts the ratio of probabilities $\pi_\theta(s_i, a_i)$ and $\pi_{\bar{\theta}}(s_i, a_i)$ for every state-action pair $(s_i, a_i)$ in the buffer as follows to attain the goal, using a constant $\epsilon_{clip}$:

$$1 - \epsilon_{clip} < \frac{\pi_\theta(s_i, a_i)}{\pi_{\bar{\theta}}(s_i, a_i)} < 1 + \epsilon_{clip}. \tag{5}$$

## 4  Approach

In this section, we discuss how we can use analytical gradients in the PPO framework while considering their variance and biases. We start our discussion with the definition of $\alpha$-policy.

### 4.1  $\alpha$-Policy

With the analytical gradient of advantage with respect to an action $(\nabla_a A_{\pi_{\bar{\theta}}}(s, a))$ from our differentiable environments, we can define a new class of policies $\pi_\alpha$, parameterized by $\alpha$, as follows.

**Definition 4.1** *Given current policy $\pi_{\bar{\theta}}$, we define its $\alpha$-policy as follows:*

$$\pi_\alpha(s, \tilde{a}) = \begin{cases} \frac{\pi_{\bar{\theta}}(s, a)}{|\det(I + \alpha \nabla_a^2 A_{\pi_{\bar{\theta}}}(s, a))|} & if \quad \exists a \ s.t. \ \tilde{a} = f(a) = a + \alpha \cdot \nabla_a A_{\pi_{\bar{\theta}}}(s, a) \\ 0 & else \end{cases}, \tag{6}$$

$$where \ constant \ |\alpha| < \frac{1}{\max_{(s,a)} |\lambda_1(s, a)|}.$$

*Here $\lambda_1(s, a)$ represents the minimum eigenvalue of $\nabla_a^2 A_{\pi_{\bar{\theta}}}(s, a)$.*

**Lemma 4.2** *Mapping $f$ is injective, and for an arbitrary open set of action $T \subset A$, $\pi_\alpha(s, \cdot)$ selects a set of action $\tilde{T} = f(A)$ with the same probability that $\pi_{\bar{\theta}}(s, \cdot)$ selects $A$.*

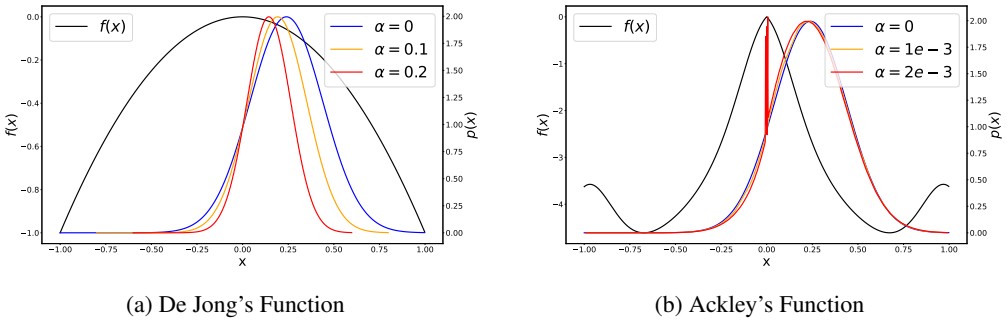

(a) De Jong's Function          (b) Ackley's Function

Figure 1: $\alpha$-policies for 1-dimensional (a) De Jong's function and (b) Ackley's function. The original policy is rendered in blue, and alpha policies for different $\alpha$ values are rendered in orange and red. Those $\alpha$-policies are obtained from analytical gradients of the given function $f(x)$ rendered in black. Note that Ackley's $\alpha$-policy becomes invalid when $\alpha = 2 \cdot 10^{-3}$, because of singularities near 0.

**Proof:** See Appendix 7.1.2.

In Figure 1, we provide renderings of $\alpha$-policies for different $\alpha$s when we use 1-dimensional De Jong's function and Ackley's function [Molga and Smutnicki, 2005] as our target functions. The illustration provides us an intuition that when $\alpha$ is sufficiently small, $\pi_\alpha$ can be seen as a policy that selects slightly better action than $\pi_{\bar\theta}$ with the same probability. In fact, we can prove that $\pi_\alpha$ indeed gives us better estimated expected return in Equation 4 when $\alpha > 0$ is sufficiently small.

**Proposition 4.3** *If $|\alpha| \ll 1$,*

$$
\begin{aligned}
L_{\pi_{\bar\theta}}(\pi_\alpha) - \eta(\pi_{\bar\theta}) = O(|\alpha|) \text{ when } \alpha > 0, \\
\eta(\pi_{\bar\theta}) - L_{\pi_{\bar\theta}}(\pi_\alpha) = O(|\alpha|) \text{ when } \alpha < 0,
\end{aligned}
\tag{7}
$$

*where $L_{\pi_{\bar\theta}}$ denotes estimated expected return defined in Equation 4.*

**Proof:** See Appendix 7.1.3.

This proposition tells us that $\alpha$-policy fits into PPO framework seamlessly, and thus if we update our policy towards $\alpha$-policy, it does not contradict the PPO's objective. However, since there is a second order derivative $\nabla_a^2 A_{\pi_{\bar\theta}}(s, a)$ in the definition, it is unclear how we can update our policy $\pi_\theta$ towards $\pi_\alpha$. In the next section, we show how we can do this and how it is related to RP gradient.

### 4.2  Relationship between RP gradient and $\alpha$-policy

To show how we can approximate $\alpha$-policy, for a deterministic function $g_{\bar\theta}$ such that $\pi_{\bar\theta} \triangleq g_{\bar\theta}$, we define a new function $g_\alpha$ as follows, and provide a Lemma about it.

$$
g_\alpha(s, \epsilon) = a + \alpha \cdot \nabla_a A_{\pi_{\bar\theta}}(s, a), \text{ where } a = g_{\bar\theta}(s, \epsilon).
\tag{8}
$$

**Lemma 4.4** *When $a = g_{\bar\theta}(s, \epsilon)$,*

$$
\det(\frac{\partial g_\alpha(s, \epsilon)}{\partial \epsilon}) \cdot \det(\frac{\partial g_{\bar\theta}(s, \epsilon)}{\partial \epsilon})^{-1} = \det(I + \alpha \cdot \nabla_a^2 A_{\pi_{\bar\theta}}(s, a)).
\tag{9}
$$

**Proof:** See Appendix 7.1.4.

Note that we can update $g_\theta$ to approximate $g_\alpha$ by minimizing the following loss.

$$
L(\theta) = \mathbb{E}_{s_0, \epsilon_0, \dots \sim q}\left[ ||g_\theta(s_t, \epsilon_t) - g_\alpha(s_t, \epsilon_t)||^2 \right].
\tag{10}
$$

In fact, it turns out that minimizing Equation 10 leads our policy $\pi_\theta$ to approximate $\pi_\alpha$, because of the following Proposition.

**Proposition 4.5** *If $\pi_{\bar\theta} \triangleq g_{\bar\theta}$, for $\alpha$ that satisfies the constraint in Definition 4.1, $\pi_\alpha \triangleq g_\alpha$.*

**Proof:** See Appendix 7.1.6.

Note that we can use different advantage function formulations for $A$ in Equation 8, such as GAE [Schulman et al., 2015b] that we use in this work. However, let us we consider the following advantage function $\hat{A}$ to see the relationship between RP gradient and $\alpha$-policy.

$$\hat{A}_{\pi_\theta}(s_t, a_t) = \frac{1}{2}\mathbb{E}_{s_t, a_t, \ldots \sim \pi_\theta}\Big[\sum_{k=t}^{\infty} \gamma^k r(s_k, a_k)\Big], \tag{11}$$

Then, the following Lemma holds.

**Lemma 4.6** *When $\alpha = 1$, if we define $g_\alpha$ with $\hat{A}$, the RP gradient corresponds to $\frac{\partial L}{\partial \theta}$ at $\theta = \bar{\theta}$.*

**Proof:** See Appendix 7.1.5.

This Lemma tells us that we can understand the RP gradient as the very first gradient we get when we update our policy toward $\pi_\alpha$ by minimizing Equation 10. In other words, $\alpha$-policy is not only a superior policy than the original one in the PPO's point of view (Proposition 4.3), but also a target policy for RP gradient (Lemma 4.6). Therefore, we conclude that $\alpha$-policy is a bridge that connects these two different policy update regimes.

## 4.3 Algorithm

Our algorithm is mainly comprised of 3 parts: (1) Update $\pi_\theta$ to $\pi_\alpha$ by minimizing Equation 10, (2) Adjust $\alpha$ for next iteration, and (3) Update $\pi_\theta$ by maximizing Equation 4. In this section, we discuss details about Step 2 and 3, and provide the outline of our algorithm.

### 4.3.1 Step 2: Adjusting $\alpha$

In Definition 4.1, we can observe that $\alpha$ controls the influence of analytical gradients in policy updates. If $\alpha = 0$, it is trivial that $\pi_\alpha = \pi_{\bar{\theta}}$, which means that analytical gradients are ignored. In contrast, as $\alpha$ gets bigger, analytical gradients play bigger roles, and $\pi_\alpha$ becomes increasingly different from $\pi_{\bar{\theta}}$. Then, our mission is to find a well-balanced $\alpha$ value. Rather than setting it as a fixed hyperparameter, we opt to adjust it adaptively. We first present the following variance and bias criteria to control $\alpha$, as it is commonly used to assess the quality of gradients [Parmas et al., 2018, Suh et al., 2022].

**Variance** After we approximate $\pi_\alpha$ in Step 1), we can estimate $\det(I + \alpha\nabla_a^2 A_{\pi_{\bar{\theta}}}(s, a))$ for every state-action pair in our buffer using Lemma 4.4, and adjust $\alpha$ to bound the estimated values in a certain range, $[1 - \delta, 1 + \delta]$. There are two reasons behind this strategy.

- As shown in Definition 4.1, we have to keep $\alpha$ small enough so that $\pi_\alpha$ does not become invalid policy. By keeping $\det(I + \alpha\nabla_a^2 A_{\pi_{\bar{\theta}}}(s, a))$ larger than $1 - \delta$ to keep its positiveness, we guarantee that our policy is not updated towards an invalid policy.
- Note that $\det(I + \alpha\nabla_a^2 A_{\pi_{\bar{\theta}}}(s, a)) = \Pi_{i=1}^n(1 + \alpha\lambda_i)$, where $\lambda_i$ is the $i$-th eigenvalue of $\nabla_a^2 A_{\pi_{\bar{\theta}}}(s, a)$. By constraining this value near 1 by adjusting $\alpha$, we can guarantee stable policy updates even when some eigenvalues of $\nabla_a^2 A_{\pi_{\bar{\theta}}}(s, a)$ have huge magnitudes, which is related to a high variance RP gradient. In Appendix 7.3.1, we provide empirical evidence that this estimation is related to the sample variance of analytical gradients.

**Bias** By Proposition 4.3, we already know that when we estimate the right term of Equation 4 after Step 1), it should be a positive value. However, if the analytical gradients were biased, this condition could not be met. In runtime, if we detect such cases, we decrease $\alpha$.

In addition, we use one more criterion to adjust $\alpha$, which we call the out-of-range ratio. This criterion was designed to promise the minimum amount of policy update by PPO.

**Out-of-range-ratio** We define out-of-range-ratio as follows,

$$\text{out-of-range-ratio} = \frac{1}{N}\sum_{i=1}^{N}\mathbb{I}(|\frac{\pi_\theta(s_i, a_i)}{\pi_{\bar{\theta}}(s_i, a_i)} - 1| > \epsilon_{clip}), \tag{12}$$

where $N$ is the size of the buffer, $\mathbb{I}$ is the indicator function, and $\epsilon_{clip}$ is the constant from Equation 5. We assess this ratio after the policy update in Step 1), allowing its value to be non-zero when $\alpha > 0$. A non-negligible value of this ratio indicates a potential violation of the PPO's restriction outlined in Equation 5, thereby compromising the effectiveness of PPO. Further elaboration can be found in Appendix 7.3.2. Therefore, when this ratio exceeds a predetermined threshold, we subsequently reduce the value of $\alpha$.

To sum it up, our strategy to control $\alpha$ aims at decreasing it when analytical gradients exhibit high variance or bias, or when there is little room for PPO updates. This results in greater dependence on PPO, which can be thought of as a safeguard in our approach. Otherwise, we increase $\alpha$, since we can expect higher expected return with a greater $\alpha$ as shown in Proposition 4.3.

### 4.3.2 Step 3: PPO Update

Since we do policy updates based on PPO after the update based on $\pi_\alpha$, there is a possibility that the PPO update can "revert" the update from Step 1). To avoid such situations, we propose to use the following $\pi_h$ in the place of $\pi_{\bar\theta}$ in Equation 5:

$$\pi_h(s,a) = \frac{1}{2}(\pi_{\bar\theta}(s,a) + \pi_\alpha(s,a)).$$

By using this virtual policy $\pi_h$, we can restrict the PPO update to be done near both $\pi_{\bar\theta}$ and $\pi_\alpha$. See Appendix 7.3.2 for more details.

### 4.3.3 Pseudocode

In Algorithm 1, we present pseudocode that illustrates the outline of our algorithm, GI-PPO. There are five hyperparameters in the algorithm, which are $\alpha_0, \beta, \delta_{det}, \delta_{oorr}$, and $\max(\alpha)$. We present specific values for these hyperparameters for our experiments in Appendix 7.4.2.

---

**Algorithm 1** GI-PPO

---

$\alpha \leftarrow \alpha_0$, Initial value
$\beta \leftarrow$ Constant multiplier larger than 1 for $\alpha$
$\delta_{det}, \delta_{oorr} \leftarrow$ Constant thresholds
$B \leftarrow$ Experience buffer
**while** *Training not ended* **do**
    Clear $B$
    **while** *Not collected enough experience* **do**
        Collect experience $\{s_t, \epsilon_t, a_t, r_t, s_{t+1}\} \rightarrow B$
    **end**
    Estimate advantage $A$ for every $(s_i, a_i)$ in $B$ using Eq. 14
    Estimate advantage gradient $\frac{\partial A}{\partial a}$ for every $(s_i, a_i)$ in $B$ using Eq. 15
    For current $\alpha$, approximate $\alpha$-policy by minimizing loss in Equation 10
    // **Variance**
    For each $\epsilon_i$ and its corresponding state-action pair $(s_i, a_i)$, estimate $\det(I + \alpha \cdot \nabla_a^2 A_{\pi_{\bar\theta}}(s,a))$ by Lemma 4.4 and get its sample minimum ($\psi_{min}$) and maximum ($\psi_{max}$)
    // **Bias**
    Evaluate expected additional return in Equation 16 with our current policy to get $R_\alpha$
    // **Out-of-range-ratio**
    Evaluate out-of-range-ratio in Equation 12 with our current policy to get $R_{oorr}$
    **if** $\psi_{min} < 1 - \delta_{det}$ *or* $\psi_{max} > 1 + \delta_{det}$ *or* $R_\alpha < 0$ *or* $R_{oorr} > \delta_{oorr}$ **then**
        $\alpha = \alpha/\beta$
    **end**
    **else**
        $\alpha = \alpha \times \beta$
    **end**
    $\alpha = clip(\alpha, 0, \max(\alpha))$
    Do PPO update by maximizing the surrogate loss in Equation 18
**end**

---

Table 1: Average maximum reward (↑) for function optimization problems.

| Problem | LR | RP | PPO | LR+RP | GI-PPO |
|---|---|---|---|---|---|
| Dejong (1) | $-1.24*10^{-6}$ | $-1.42*10^{-8}$ | $-5.21*10^{-5}$ | $-6.36*10^{-8}$ | $\mathbf{-3.84*10^{-10}}$ |
| Dejong (64) | $-0.0007$ | $\mathbf{-9.28*10^{-7}}$ | $-0.0011$ | $-3.05*10^{-6}$ | $-1.04*10^{-6}$ |
| Ackley (1) | $-1.2772$ | $-0.4821$ | $-0.2489$ | $-1.2255$ | $\mathbf{-0.0005}$ |
| Ackley (64) | $-0.6378$ | $-0.0089$ | $-0.1376$ | $-0.0326$ | $\mathbf{-0.0036}$ |

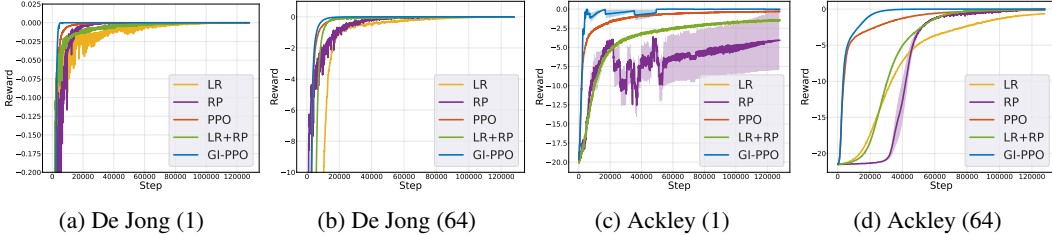

| (a) De Jong (1) | (b) De Jong (64) | (c) Ackley (1) | (d) Ackley (64) |
|---|---|---|---|

Figure 2: Optimization curves for Dejong's and Ackley's function of dimension 1 and 64.

## 5 Experimental Results

In this section, we present experimental results that show our method's efficacy for various optimization and complex control problems. To validate our approach, we tested various baseline methods on the environments that we use. The methods are as follows:

- LR: Policy gradient method based on LR gradient.
- RP: Policy gradient method based on RP gradient. For physics and traffic problems, we adopted a truncated time window of [Xu et al., 2022] to reduce variance.
- PPO: Proximal Policy Optimization [Schulman et al., 2017].
- LR+RP: Policy gradient method based on interpolation between LR and RP gradient using sample variance [Parmas et al., 2018].
- PE: Policy enhancement scheme of [Qiao et al., 2021], for physics environments only.
- GI-PPO: Our method based on Section 4.3.

Please refer Appendix 7.4 to learn about implementation details of these baseline methods, network architectures, and hyperparameters that we used for all of the experiments.

We have implemented our learning method using PyTorch 1.9 Paszke et al. [2019]. As for hardware, all experiments were run with an Intel® Xeon® W-2255 CPU @ 3.70GHz, one NVIDIA RTX A5000 graphics card, and 16 GB of memory. Experiment results are averaged across 5 random seeds.

### 5.1 Function Optimization Problems

We start with how different learning methods find the maximum point for a given analytical function. We ran these experiments because RL algorithms can be thought of as function optimizers through function sampling [Williams and Peng, 1989]. These problems are 1-step optimization problems, where agents guess an optimal point as an action and get the negative function value as the reward.

As we used for rendering $\alpha$-policies in Figure 1, we use De Jong's function and Ackley's function for comparison, as they are popular functions for testing numerical optimization algorithms [Molga and Smutnicki, 2005]. Please see Appendix 7.5.1 for the details about the function definitions. In this setting, the optimum occurs at the point $x = 0$ for both of the problems. However, their loss landscapes are quite different from one another—De Jong's function has a very smooth landscape that only has one local (global) maximum, while Ackley's function has a very rugged landscape with multiple local maxima. Therefore, De Jong's function represents environments where RP gradients have lower variance, while Ackley's function represents the opposite.

Table 1 shows that **GI-PPO finds far better optimums than the other methods for most of the functions**, except 64-dimensional Dejong's function, where the RP method found slightly better

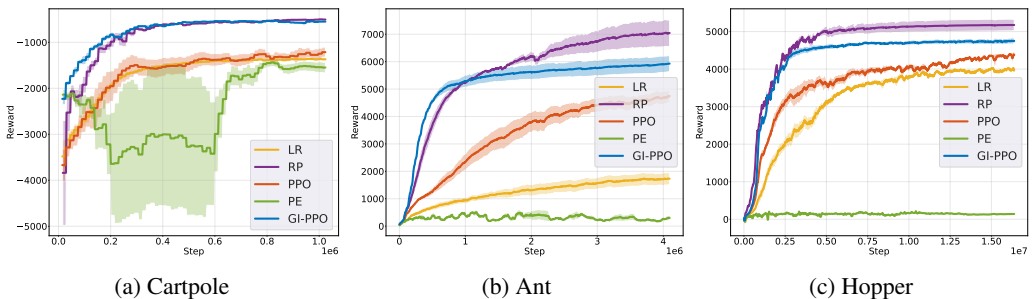

|             |             |             |
|:-----------:|:-----------:|:-----------:|
| (a) Cartpole | (b) Ant | (c) Hopper |

Figure 3: Learning graphs for 3 problems in differentiable physics simulation: Cartpole, Ant, and Hopper [Xu et al., 2022]. In these environments, RP gradients already exhibit lower variance without much bias than LR gradients, because of variance reduction scheme of [Xu et al., 2022].

optimum than GI-PPO. Optimization curves in Figure 2 also prove that our method converges to the optimum faster than the other methods. Interestingly, the RP method achieved better results than the LR and LR+RP methods for all the functions, which contradicts our assumption that it would not work well for Ackley's function. However, we found out that even though the RP method takes some time to find the optimum than the other methods as shown in Figure 2, when it approaches near-optimum, it converges fast to the optimum based on analytic gradients. Since GI-PPO adaptively changes $\alpha$ to favor RP gradient near optimum, it could achieve better results than the other methods. In Appendix 7.3.3, we provide visual renderings that trace the change of $\alpha$ value during optimization to help understanding, which also shows that higher $\alpha$ is maintained for Dejong's function than Ackley's function, which aligns with our initial assumption. Also note that even though LR+RP also takes advantage of the RP method, and thus achieved a better convergence rate than the two methods when it is far from optimum, it did not work well when it comes to near-optimum.

## 5.2 Differentiable Physics Simulation

Next, we conducted experiments in differentiable physics simulations. We used Cartpole, Ant, and Hopper environments implemented by [Xu et al., 2022] for comparisons. However, since the current implementation does not support multiple backpropagations through the same computation graph for these environments, we could not test the LP+RP method for them. Instead, we tried to faithfully implement another policy enhancement (PE) strategy of [Qiao et al., 2021] as a comparison, as it showed its efficacy in differentiable physics simulations using analytical gradients.

In Figure 3, we can see that GI-PPO converged to a better policy much faster than the baseline PPO for every environment, which validates our approach. However, the RP method achieved the best results for Ant and Hopper, while GI-PPO converged to the optimal policy slightly faster for Cartpole. To explain this result, we'd like to point out that $\alpha$ is upper bounded by out-of-range-ratio in our approach (Section 4.3.1). Therefore, even if the RP gradient is very effective, we cannot fully utilize it because we have to use PPO to update the policy up to some extent as a safeguard. In the Ant environment, we could observe that it was the major bottleneck in training—in fact, GI-PPO could achieve better results than RP when we increased the ratio from 0.5 to 1.0 (Appendix 7.3.4). However, when the out-of-range ratio equals 1.0, it becomes more likely that we cannot use PPO as a safeguard, which contradicts our design intention. When it comes to the Hopper environment, we found that the variance of proper $\alpha$ values over time was large, so our strategy could not conform to it properly. Overcoming these limitations would be an interesting future work to extend our work.

## 5.3 Traffic Problems

Here we demonstrate the effectiveness of GI-PPO in an environment where continuous and discrete operations coexist, which leads to highly biased analytical gradients. We suggest a mixed-autonomy traffic environment as a suitable benchmark, which has previously been explored as a benchmark for gradient-free RL algorithms [Wu et al., 2017, Vinitsky et al., 2018]. In this paper, we use the pace car problem, where a single autonomous pace car has to control the speed of the other vehicles via interference. The number of lanes, which represent the discontinuities in gradients, and the number of following human vehicles are different for each problem. Please see Appendix 7.5.2 for the details of this environment, and why the analytical gradients are biased.

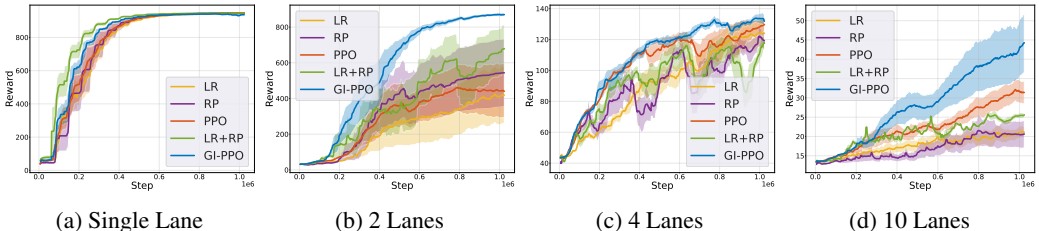

| (a) Single Lane | (b) 2 Lanes | (c) 4 Lanes | (d) 10 Lanes |

Figure 4: Learning graphs for traffic pace car problem. Each problem has different number of lanes and following human vehicles, which represent *discontinuities* and *dimension of the problem*, respectively.

In Figure 4, we can observe that GI-PPO exhibits a faster convergence rate and converges to better policy overall compared to all the other methods in 2, 4, and 10-lane environments. In a single-lane environment, however, LR+RP showed a better convergence rate than GI-PPO. Note that the RP method still performs well for 2 and 4-lane environments just as other baseline methods, even if there were multiple lanes. We assume this is because the RP method also leverages a state value function, which can provide unbiased analytical gradients in the presence of discontinuities. However, it does not perform well in the 10-lane environment, where discontinuities are excessive compared to the other environments.

This result shows that **GI-PPO can still leverage analytical gradients, even if they were highly biased for policy learning**. Even though LP+RP, which is based on a similar spirit as ours, also exhibited some improvements over LP and RP, *our method showed better results for more difficult problems because baseline PPO offers more stable learning than baseline LR*. Also, the computational cost of GI-PPO was a little more than RP, while that of LR+RP was much more expensive (Appendix 7.7).

# 6   Conclusions

We introduced a novel policy learning method that adopts analytical gradients into the PPO framework. Based on the observation that the RP gradient leads our current policy to its $\alpha$-policy, we suggested approximating it explicitly, which allows us to indirectly estimate variance and bias of analytical gradients. We suggested several criteria to detect such variance and bias and introduced an algorithm that manipulates $\alpha$ adaptively, which stands for the strength of analytical gradients in policy updates. In our experiments of diverse RL environments, we successfully showed that our method achieves much better results than the baseline PPO by adopting analytical gradients. Even though the interpolated gradient of LR and RP gradient is based on similar motivations, our method performed much better, thanks to the baseline PPO's stable learning.

**Limitations**   Despite GI-PPO's promising results, there are some remaining issues. First, even if we use a large $\alpha$, we may not fully approximate the corresponding $\alpha$-policy in a limited number of optimization steps, which may give a false signal for our strategy. We can upper bound the maximum $\alpha$ as we did here, but a more rigorous optimization process may be helpful. This is also related to improving the strategy to control $\alpha$ adaptively, which would be a promising future work.

Also, as discussed in Section 5.2, our method is tightly bound to PPO—that is, even when analytical gradients are much more useful as illustrated in differentiable physics problems, we cannot fully utilize them if they exit the PPO's bound. Even though it is more suitable for stable updates, it could result in a worse convergence rate. If we could adjust PPO's clipping range dynamically, or detect environments where RP gradients are much more reliable, we would be able to overcome this issue.

Finally, computational costs can be further reduced. To utilize analytical gradients, we need backpropagation, which usually requires more time than forward steps. This is the reason why the learning methods based on analytical gradients require longer training time than the others (Appendix 7.7). When the time window gets longer, the cost grows accordingly. However, as we have shown in our traffic experiments, our method works even when gradients are biased. Therefore, it would be an interesting research direction to see if using a very short time window for backpropagation, which produces more biased gradients, would be worthy of exploring for improved computational efficiency.

**Acknowledgements.** This research is supported in part by the ARO DURIP and IARPA HAYSTAC Grants, ARL Cooperative Agreement W911NF2120076, Dr. Barry Mersky and Capital One E-Nnovate Endowed Professorships. Yi-Ling Qiao would like to thank the support of Meta Fellowship.

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

# 7 APPENDIX

## 7.1 Proofs

### 7.1.1 Proof of Lemma 3.2

By Definition 3.1, $\pi_\theta(s, a)$ should satisfy the following condition:

$$\int_{T_\epsilon} q(\epsilon)d\epsilon = \int_{T_a} \pi_\theta(s, a)da$$
$$= \int_{T_\epsilon} \pi_\theta(s, g_\theta(s, \epsilon)) \cdot \left|\det(\frac{\partial g_\theta(s, \epsilon)}{\partial \epsilon})\right|d\epsilon \quad \text{(by change of variable)}.$$

We denote the set of continuous distributions that satisfy this condition as $P$. That is,

$$P_\theta(s, \epsilon) \in P \text{ if } \int_{T_\epsilon} q(\epsilon)d\epsilon = \int_{T_\epsilon} P_\theta(s, \epsilon) \cdot \left|\det(\frac{\partial g_\theta(s, \epsilon)}{\partial \epsilon})\right|d\epsilon.$$

Clearly, $\bar{P}_\theta(s, \epsilon) = q(\epsilon) \cdot \left|\det(\frac{\partial g_\theta(s,\epsilon)}{\partial \epsilon})\right|^{-1}$ is an element of $P$, and is well defined, because of Equation 3. In fact, $\bar{P}_\theta(s, \epsilon)$ is the only element of $P$, as the above condition should be met for any $T_\epsilon \subset \mathbb{R}^n$. That is, if there was another distribution $\tilde{P}_\theta(s, \epsilon) \in P$,

$$\int_{T_\epsilon} (\bar{P}_\theta(s, \epsilon) - \tilde{P}_\theta(s, \epsilon)) \cdot \left|\det(\frac{\partial g_\theta(s, \epsilon)}{\partial \epsilon})\right|d\epsilon = 0, \forall T_\epsilon \subset \mathbb{R}^n.$$

Since $\tilde{P}_\theta$ is different from $\bar{P}_\theta$, there exists an open set $\hat{T}_\epsilon$ where $\tilde{P}_\theta < \bar{P}_\theta$. Then

$$\int_{\hat{T}_\epsilon} (\bar{P}_\theta(s, \epsilon) - \tilde{P}_\theta(s, \epsilon)) \cdot \left|\det(\frac{\partial g_\theta(s, \epsilon)}{\partial \epsilon})\right|d\epsilon > 0,$$

which is a contradiction. Therefore, $\pi_\theta(s, a) = \pi_\theta(s, g_\theta(s, \epsilon)) = \bar{P}_\theta(s, \epsilon)$. The inverse holds trivially.

### 7.1.2 Proof of Lemma 4.2

First, we show that $f$ is an injective function by showing that the determinant of Jacobian $\frac{\partial f}{\partial a}$ is always positive:

$$\det(\frac{\partial f}{\partial a}) = \det(I + \alpha \cdot \nabla_a^2 A_{\pi_{\bar{\theta}}}(s, a))$$
$$= \Pi_{i=1}^n (1 + \alpha \cdot \lambda_i(s, a)) > 0 \quad (\because |\alpha| < \frac{1}{\max_{(s,a)}|\lambda_1(s, a)|}),$$

where $\lambda_i(s, a)$ are the eigenvalues of $\nabla_a^2 A_{\pi_{\bar{\theta}}}(s, a)$ sorted in ascending order.

Then, for an arbitrary open set of action $T \subset A$, $\pi_\alpha(s, \cdot)$ selects $\tilde{T} = f(T) \subset A$ with the same probability that the original policy $\pi_{\bar{\theta}}(s, \cdot)$ selects $T$.

$$\int_{\tilde{T}} \pi_\alpha(s, \tilde{a})d\tilde{a} = \int_T \frac{\pi_{\bar{\theta}}(s, a)}{|\det(I + \alpha\nabla_a^2 A_{\pi_{\bar{\theta}}}(s, a))|} \cdot \left|\det(\frac{\partial f}{\partial a})\right|da$$
$$= \int_T \frac{\pi_{\bar{\theta}}(s, a)}{|\det(I + \alpha\nabla_a^2 A_{\pi_{\bar{\theta}}}(s, a))|} \cdot \left|\det(I + \alpha \cdot \nabla_a^2 A_{\pi_{\bar{\theta}}}(s, a))\right|da$$
$$= \int_T \pi_{\bar{\theta}}(s, a)da.$$

Therefore, $\pi_\alpha(s, \cdot)$ is a valid probability distribution as $\pi_{\bar{\theta}}(s, \cdot)$.

### 7.1.3 Proof of Proposition 4.3

For a given state $s$, we can estimate the (approximate) expected value of the state under $\pi_\alpha$ as follows.

$$\int_{\tilde{a}} \pi_\alpha(s, \tilde{a}) A_{\pi_{\bar{\theta}}}(s, \tilde{a}) d\tilde{a}$$

$$= \int_a \frac{\pi_{\bar{\theta}}(s, a)}{|\det(I + \alpha \nabla_a^2 A_{\pi_{\bar{\theta}}}(s, a))|} A_{\pi_{\bar{\theta}}}(s, a + \alpha \nabla_a A_{\pi_{\bar{\theta}}}(s, a)) |\det(I + \alpha \nabla_a^2 A_{\pi_{\bar{\theta}}}(s, a))| da$$

$$\approx \int_a \pi_{\bar{\theta}}(s, a) \left[ A_{\pi_{\bar{\theta}}}(s, a) + \alpha ||\nabla_a A_{\pi_{\bar{\theta}}}(s, a)||^2 \right] da$$

$$= \int_a \pi_{\bar{\theta}}(s, a) A_{\pi_{\bar{\theta}}}(s, a) da + \alpha \int_a \pi_{\bar{\theta}}(s, a) ||\nabla_a A_{\pi_{\bar{\theta}}}(s, a)||^2 da.$$

Therefore, we can see the following holds:

$$L_{\pi_{\bar{\theta}}}(\pi_\alpha) = \int_s \rho_{\pi_{\bar{\theta}}}(s) \int_{\tilde{a}} \pi_\alpha(s, \tilde{a}) A_{\pi_{\bar{\theta}}}(s, \tilde{a}) d\tilde{a}$$

$$\approx \eta(\pi_{\bar{\theta}}) + \alpha \int_s \rho_{\pi_{\bar{\theta}}}(s) \int_a \pi_{\bar{\theta}}(s, a) ||\nabla_a A_{\pi_{\bar{\theta}}}(s, a)||^2 da.$$

Since $\pi_{\bar{\theta}}(s, a)$ and $||\nabla_a A_{\pi_{\bar{\theta}}}(s, a)||^2$ are positive, $L_{\pi_{\bar{\theta}}}(\pi_\alpha)$ is greater or equal to $\eta(\pi_{\bar{\theta}})$ when $\alpha > 0$. On the contrary, when $\alpha < 0$, $L_{\pi_{\bar{\theta}}}(\pi_\alpha)$ is smaller or equal to $\eta(\pi_{\bar{\theta}})$. Since this argument builds upon local approximation, it holds only when $|\alpha| \ll 1$.

### 7.1.4 Proof of Lemma 4.4

When $a = g_{\bar{\theta}}(s, \epsilon)$,

$$g_\alpha(s, \epsilon) = g_{\bar{\theta}}(s, \epsilon) + \alpha \cdot \nabla_{g_{\bar{\theta}}(s, \epsilon)} A_{\pi_{\bar{\theta}}}(s, g_{\bar{\theta}}(s, \epsilon))$$

$$\Rightarrow \frac{\partial g_\alpha(s, \epsilon)}{\partial \epsilon} = \frac{\partial g_{\bar{\theta}}(s, \epsilon)}{\partial \epsilon} + \alpha \cdot \nabla_{g_{\bar{\theta}}(s, \epsilon)}^2 A_{\pi_{\bar{\theta}}}(s, g_{\bar{\theta}}(s, \epsilon)) \cdot \frac{\partial g_{\bar{\theta}}(s, \epsilon)}{\partial \epsilon}$$

$$= \left( I + \alpha \cdot \nabla_{g_{\bar{\theta}}(s, \epsilon)}^2 A_{\pi_{\bar{\theta}}}(s, g_{\bar{\theta}}(s, \epsilon)) \right) \cdot \frac{\partial g_{\bar{\theta}}(s, \epsilon)}{\partial \epsilon}$$

$$\Rightarrow \det \left( \frac{\partial g_\alpha(s, \epsilon)}{\partial \epsilon} \right) = \det \left( I + \alpha \cdot \nabla_{g_{\bar{\theta}}(s, \epsilon)}^2 A_{\pi_{\bar{\theta}}}(s, g_{\bar{\theta}}(s, \epsilon)) \right) \cdot \det \left( \frac{\partial g_{\bar{\theta}}(s, \epsilon)}{\partial \epsilon} \right)$$

$$= \det \left( I + \alpha \cdot \nabla_a^2 A_{\pi_{\bar{\theta}}}(s, a) \right) \cdot \det \left( \frac{\partial g_{\bar{\theta}}(s, \epsilon)}{\partial \epsilon} \right).$$

Since $\left| \det \left( \frac{\partial g_{\bar{\theta}}(s, \epsilon)}{\partial \epsilon} \right) \right| > 0$ (Equation 3), we can divide both sides of the above equation with $\det \left( \frac{\partial g_{\bar{\theta}}(s, \epsilon)}{\partial \epsilon} \right)$ and prove Lemma 4.4.

### 7.1.5 Proof of Lemma 4.6

From Equation 11,

$$\nabla_a \hat{A}_{\pi_\theta}(s_t, a_t) = \frac{1}{2} \mathbb{E}_{s_t, a_t, \dots \sim \pi_\theta} \left[ \sum_{k=t}^{\infty} \gamma^k \frac{\partial r(s_k, a_k)}{\partial a_t} \right],$$

which can be rewritten using $g_\theta$ as follows.

$$\nabla_a \hat{A}_{\pi_\theta}(s_t, a_t) = \frac{1}{2} \mathbb{E}_{s_t, \epsilon_t, \dots \sim q} \left[ \sum_{k=t}^{\infty} \gamma^k \frac{\partial r(s_k, g_\theta(s_t, \epsilon_t))}{\partial g_\theta(s_t, \epsilon_t)} \right]. \tag{13}$$

Then, by differentiating Equation 10 with respect to $\theta$ after plugging in $\hat{A}$, we can get following relationship:

$$\frac{\partial L(\theta)}{\partial \theta} = \mathbb{E}_{s_0,\epsilon_0,\ldots,\sim q}\Big[\sum_{t=0}^{\infty} 2 \cdot \frac{\partial g_\theta(s_t,\epsilon_t)}{\partial \theta} \cdot (g_\theta(s_t,\epsilon_t) - g_\alpha(s_t,\epsilon_t))\Big]$$

$$= \mathbb{E}_{s_0,\epsilon_0,\ldots,\sim q}\Big[\sum_{t=0}^{\infty} 2 \cdot \frac{\partial g_\theta(s_t,\epsilon_t)}{\partial \theta} \cdot (g_\theta(s_t,\epsilon_t) - (g_\theta(s_t,\epsilon_t) + \nabla_a \hat{A}_{\pi_\theta}(s_t,a_t)))\Big]$$

$$(\because \alpha = 1)$$

$$= \mathbb{E}_{s_0,\epsilon_0,\ldots,\sim q}\Big[\sum_{t=0}^{\infty} -2 \cdot \frac{\partial g_\theta(s_t,\epsilon_t)}{\partial \theta} \cdot \nabla_a \hat{A}_{\pi_\theta}(s_t,a_t)\Big]$$

$$= \mathbb{E}_{s_0,\epsilon_0,\ldots,\sim q}\Big[\sum_{t=0}^{\infty} -2 \cdot \frac{\partial g_\theta(s_t,\epsilon_t)}{\partial \theta} \cdot \frac{1}{2} \cdot \sum_{k=t}^{\infty} \gamma^k \frac{\partial r(s_k, g_\theta(s_t,\epsilon_t))}{\partial g_\theta(s_t,\epsilon_t)}\Big]$$

$$(\because \text{Equation 13})$$

$$= \mathbb{E}_{s_0,\epsilon_0,\ldots,\sim q}\Big[\sum_{t=0}^{\infty}\sum_{k=t}^{\infty} -\gamma^k \frac{\partial g_\theta(s_t,\epsilon_t)}{\partial \theta} \cdot \frac{\partial r(s_k, g_\theta(s_t,\epsilon_t))}{\partial g_\theta(s_t,\epsilon_t)}\Big],$$

which equals to RP gradient in Appendix 7.2.2, but with different sign. However, they turn out to be the same because we minimize $L(\theta)$, but maximize $\eta(\pi_\theta)$. Therefore, we can say that we gain RP gradient as the first gradient when we minimize Equation 10 for particular advantage function $\hat{A}$ and $\alpha = 1$.

### 7.1.6 Proof of Proposition 4.5

By Definition 4.1 and Lemma 4.4,

$$\frac{\pi_\alpha(s,\tilde{\alpha})}{\pi_{\bar{\theta}}(s,a)} = \frac{1}{|\det(I + \alpha \cdot \nabla_a^2 A_{\pi_{\bar{\theta}}}(s,a))|}$$

$$= |\det(\frac{\partial g_{\bar{\theta}}(s,\epsilon)}{\partial \epsilon})| \cdot |\det(\frac{\partial g_\alpha(s,\epsilon)}{\partial \epsilon})|^{-1},$$

where $a = g_\theta(s,\epsilon)$, and thus $\tilde{a} = g_\alpha(s,\epsilon)$. Since $\pi_{\bar{\theta}} \triangleq g_{\bar{\theta}}$, following holds:

$$\pi_\alpha(s,\tilde{\alpha}) = \pi_{\bar{\theta}}(s,a) \cdot |\det(\frac{\partial g_{\bar{\theta}}(s,\epsilon)}{\partial \epsilon})| \cdot |\det(\frac{\partial g_\alpha(s,\epsilon)}{\partial \epsilon})|^{-1}$$

$$= q(\epsilon) \cdot |\det(\frac{\partial g_{\bar{\theta}}(s,\epsilon)}{\partial \epsilon})|^{-1} \cdot |\det(\frac{\partial g_{\bar{\theta}}(s,\epsilon)}{\partial \epsilon})| \cdot |\det(\frac{\partial g_\alpha(s,\epsilon)}{\partial \epsilon})|^{-1}$$

$$(\because \text{Lemma 3.2})$$

$$= q(\epsilon) \cdot |\det(\frac{\partial g_\alpha(s,\epsilon)}{\partial \epsilon})|^{-1},$$

which implies $\pi_\alpha \triangleq g_\alpha$.

## 7.2 Formulations

### 7.2.1 Analytical Gradient of Generalized Advantage Estimator (GAE)

GAE [Schulman et al., 2015b] has been widely used in many RL implementations [Schulman et al., 2017, Raffin et al., 2021, Makoviichuk and Makoviychuk, 2022] to estimate advantages. GAE finds a balance between variance and bias of the advantage estimation by computing the exponentially-weighted average of the TD residual terms ($\delta_t^V$) Sutton et al. [1998], which are defined as follows:

$$\delta_t^V = r_t + \gamma V(s_{t+1}) - V(s_t).$$

Then, GAE can be formulated as follows:

$$A_t^{GAE}(\gamma, \lambda) = \sum_{k=0}^{\infty} (\gamma\lambda)^k \delta_{t+k}^V. \tag{14}$$

We can compute the gradients for these terms as:

$$
\begin{aligned}
\frac{\partial A_t^{GAE}}{\partial a_t} &= \sum_{k=0}^{\infty} (\gamma\lambda)^k \frac{\partial \delta_{t+k}^V}{\partial a_t} \\
&= (\frac{1}{\gamma\lambda})^t \sum_{k=0}^{\infty} (\gamma\lambda)^{t+k} \frac{\partial \delta_{t+k}^V}{\partial a_t} \\
&= (\frac{1}{\gamma\lambda})^t \left[ \sum_{k=0}^{\infty} (\gamma\lambda)^{t+k} \frac{\partial \delta_{t+k}^V}{\partial a_t} + \sum_{k=0}^{t-1} (\gamma\lambda)^k \frac{\partial \delta_k^V}{\partial a_t} \right] \\
&\quad (\because \frac{\partial \delta_k^V}{\partial a_t} = 0 \text{ for } k < t) \\
&= (\frac{1}{\gamma\lambda})^t \sum_{k=0}^{\infty} (\gamma\lambda)^k \frac{\partial \delta_k^V}{\partial a_t} \\
&= (\frac{1}{\gamma\lambda})^t \frac{\partial A_0^{GAE}}{\partial a_t}.
\end{aligned}
\tag{15}
$$

Based on this relationship, we can compute every $\frac{\partial A_t^{GAE}}{\partial a_t}$ with only one backpropagation of $A_0^{GAE}$, rather than backpropagating for every $A_t^{GAE}$.

### 7.2.2 RP Gradient Formulation

To differentiate Equation 1 with respect to $\theta$, we first rewrite Equation 1 as follows.

$$
\begin{aligned}
\eta(\pi_\theta) &= \mathbb{E}_{s_0,a_0,\ldots\sim\pi_\theta} \left[ \sum_{t=0}^{\infty} \gamma^t r(s_t, a_t) \right] \\
&= \mathbb{E}_{s_0,\epsilon_0,\ldots\sim q} \left[ \sum_{t=0}^{\infty} \gamma^t r(s_t, g_\theta(s_t, \epsilon_t)) \right].
\end{aligned}
$$

Then, we can compute RP gradient as follows.

$$
\begin{aligned}
\frac{\partial \eta(\pi_\theta)}{\partial \theta} &= \frac{\partial}{\partial \theta} \mathbb{E}_{s_0,\epsilon_0,\ldots\sim q} \left[ \sum_{t=0}^{\infty} \gamma^t r(s_t, g_\theta(s_t, \epsilon_t)) \right] \\
&= \mathbb{E}_{s_0,\epsilon_0,\ldots\sim q} \left[ \frac{\partial}{\partial \theta} \sum_{t=0}^{\infty} \gamma^t r(s_t, g_\theta(s_t, \epsilon_t)) \right] \\
&= \mathbb{E}_{s_0,\epsilon_0,\ldots\sim q} \left[ \sum_{t=0}^{\infty} \frac{\partial g_\theta(s_t, \epsilon_t)}{\partial \theta} \sum_{k=t}^{\infty} \gamma^k \frac{\partial r(s_k, g_\theta(s_k, \epsilon_k))}{\partial g_\theta(s_t, \epsilon_t)} \right] \\
&\quad (\because \frac{\partial r(s_k, g_\theta(s_k, \epsilon_k))}{\partial g_\theta(s_t, \epsilon_t)} \neq 0 \text{ only when } k \geq t.) \\
&= \mathbb{E}_{s_0,\epsilon_0,\ldots\sim q} \left[ \sum_{t=0}^{\infty} \sum_{k=t}^{\infty} \gamma^k \frac{\partial g_\theta(s_t, \epsilon_t)}{\partial \theta} \frac{\partial r(s_k, g_\theta(s_k, \epsilon_k))}{\partial g_\theta(s_t, \epsilon_t)} \right] \\
&= \mathbb{E}_{s_0,\epsilon_0,\ldots\sim q} \left[ \sum_{t=0}^{\infty} \sum_{k=t}^{\infty} \gamma^k \frac{\partial g_\theta(s_t, \epsilon_t)}{\partial \theta} \frac{\partial r(s_k, a_k)}{\partial a_t} \right].
\end{aligned}
$$

We can estimate this RP gradient using Monte Carlo sampling and terms in Equation 2, and use it for gradient ascent to maximize Equation 1.

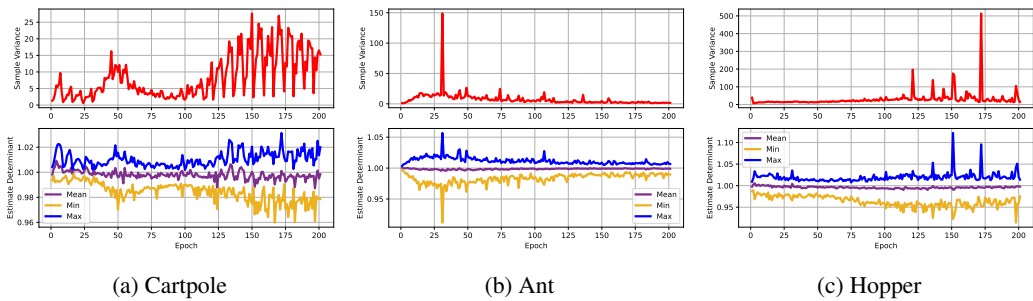

(a) Cartpole           (b) Ant           (c) Hopper

Figure 5: The first row shows the sample variance of analytical gradients ($\frac{\partial A}{\partial a}$), and the second row shows the statistics of estimates of $\det(I + \alpha\nabla_a^2 A_{\pi_{\bar{\theta}}}(s, a))$ for every state-action pair in the buffer after each epoch of training in 3 different differentiable physics environments (Cartpole, Ant, and Hopper). We used a fixed $\alpha = 10^{-2}$. Note that the minimum and maximum estimates in the second row exhibit a clear tendency to deviate from 1 more as the sample variance increases.

### 7.2.3 Estimator for Equation 4

Since $\eta(\pi_{\bar{\theta}})$ does not depend on $\theta$, we can ignore the term and rewrite our loss function in Equation 4 using expectation as follows:

$$L_{\pi_{\bar{\theta}}}(\pi_\theta) = \int_s \rho_{\pi_{\bar{\theta}}}(s) \int_a \pi_\theta(s, a) A_{\pi_{\bar{\theta}}}(s, a)$$
$$= \mathbb{E}_{s\sim\rho_{\pi_{\bar{\theta}}}, a\sim\pi_\theta(s,\cdot)}\Big[A_{\pi_{\bar{\theta}}}(s, a)\Big].$$

However, note that we did not collect trajectories, or experience buffer, using $\pi_\theta$. Therefore, we use another importance sampling function $q(s, a)$ [Schulman et al., 2015a].

$$L_{\pi_{\bar{\theta}}}(\pi_\theta) = \mathbb{E}_{s\sim\rho_{\pi_{\bar{\theta}}}, a\sim q(s,\cdot)}\Big[\frac{\pi_\theta(s, a)}{q(s, a)} A_{\pi_{\bar{\theta}}}(s, a)\Big].$$

Then, we can estimate this value using Monte Carlo estimation as follows, where $B$ is the experience buffer of size $N$, which stores state ($s_i$), action ($a_i$), and their corresponding advantage value ($A_{\pi_{\bar{\theta}}}(s_i, a_i)$) obtained by following policy $\pi_{\bar{\theta}}$:

$$L_{\pi_{\bar{\theta}}}(\pi_\theta) \approx \frac{1}{N}\sum_{i=1}^{N} \frac{\pi_\theta(s_i, a_i)}{\pi_{\bar{\theta}}(s_i, a_i)} A_{\pi_{\bar{\theta}}}(s_i, a_i). \tag{16}$$

Note that we used $\pi_{\bar{\theta}}$ in the place of $q$, as it is the most natural importance sampling function we can use [Schulman et al., 2015a, 2017].

### 7.3 Algorithm

Here we present the details of our algorithm in Section 4.3. To be specific, we present 1) an empirical evidence to validate our strategy to control $\alpha$, 2) the exact loss function that we use in PPO update, and 3) provide the change of $\alpha$ during real training to help understanding.

#### 7.3.1 Empirical evidence

In Section 4.3, we argued that the estimate of $\det(I+\alpha\nabla_a^2 A_{\pi_{\bar{\theta}}}(s, a))$ is related to the sample variance of analytical gradients, and we can get the estimate using Lemma 4.4. In Figure 5, we empirically prove the validity of this argument in the differentiable physics environments used in Section 5.2. In the illustration, we can observe a clear relationship between the sample variance of analytical gradients and the statistics of our estimates. Therefore, we can use these estimates as a cue to control $\alpha$ based on the variance of analytical gradients.

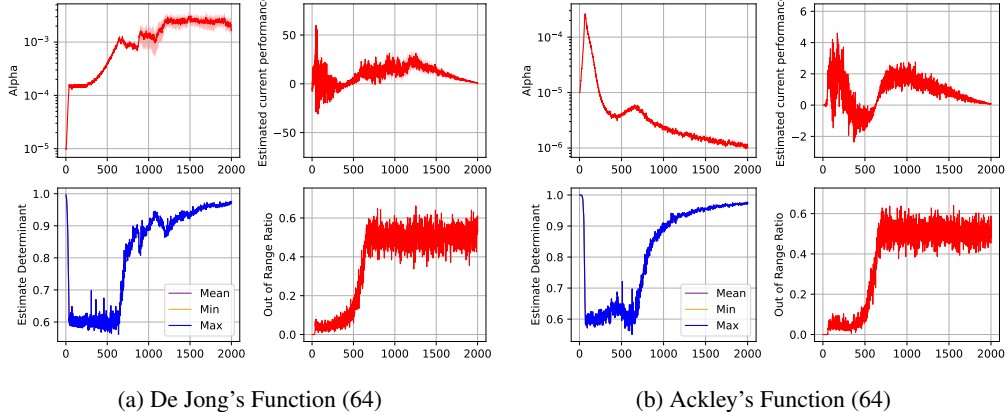

(a) De Jong's Function (64)  (b) Ackley's Function (64)

Figure 6: Change of $\alpha$ during solving function optimization problems in Section 5.1. Since these problems are one-step problems, and observations are all the same, there is no difference between mean, min, and max of estimated determinant.

### 7.3.2 Loss function for PPO Update

As discussed in Section 3.3.2, PPO finds a better policy in the proximity of the original policy by restricting the ratio of probabilities $\pi_\theta(s_i, a_i)$ and $\pi_{\bar\theta}(s_i, a_i)$, as shown in Equation 5. In fact, PPO uses following surrogate loss function, which is a modified version of Equation 16, to mandate this:

$$L_{PPO}(\theta) = \frac{1}{N} \sum_{i=1}^N \min(\frac{\pi_\theta(s_i, a_i)}{\pi_{\bar\theta}(s_i, a_i)} A_{\pi_{\bar\theta}}(s_i, a_i), clip(\frac{\pi_\theta(s_i, a_i)}{\pi_{\bar\theta}(s_i, a_i)}, 1 - \epsilon_{clip}, 1 + \epsilon_{clip}) A_{\pi_{\bar\theta}}(s_i, a_i)).$$

(17)

Note that if out-of-range-ratio defined in Equation 12 is already 1.0, which corresponds to an extreme case, the gradient of $L_{PPO}(\theta)$ with respect to $\theta$ is 0. Therefore, PPO does not contribute to the policy update at all in this case. It justifies our decision to upper bound out-of-range-ratio by restricting $\alpha$.

Also note that we proposed to use a virtual policy $\pi_h$ instead of $\pi_{\bar\theta}$ in Section 4.3.2 to restrict PPO update to be done around $\pi_\alpha$. This changes the importance sampling function in Equation 17 as follows, which corresponds to our final surrogate loss function to use in PPO update:

$$L_{GIPPO}(\theta) = \frac{1}{N} \sum_{i=1}^N \min(\frac{\pi_\theta(s_i, a_i)}{\pi_h(s_i, a_i)} A_{\pi_{\bar\theta}}(s_i, a_i), clip(\frac{\pi_\theta(s_i, a_i)}{\pi_h(s_i, a_i)}, 1 - \epsilon_{clip}, 1 + \epsilon_{clip}) A_{\pi_{\bar\theta}}(s_i, a_i)).$$

(18)

### 7.3.3 Example: Function Optimization Problems (Section 5.1)

Here we present how $\alpha$ changes as we solve function optimization problems in Section 5.1 with Figure 6. There are 4 plots for each of the problems.

- Alpha: Shows change of $\alpha$ over training epoch.
- Estimated current performance: Shows expected additional return, which corresponds to Equation 16 ($R_\alpha$ in Algorithm 1).
- Estimate Determinant: Shows statistics of estimated $\det(I + \alpha \cdot \nabla_a^2 A_{\pi_{\bar\theta}}(s, a))$.
- Out of Range Ratio: Shows out-of-range-ratio in Equation 12.

For these problems, we have set $\alpha_0 = 10^{-5}, \delta_{det} = 0.4$, and $\delta_{oorr} = 0.5$.

For De Jong's function (Figure 6a), we can observe that $\alpha$ shows steady increase over time, but it is mainly upper bounded by variance criterion and out-of-range-ratio. In the end, $\alpha$ stabilizes around $10^{-3}$. In contrast, for Ackley's function (Figure 6b) we can see that $\alpha$ increases rapidly at first, but it

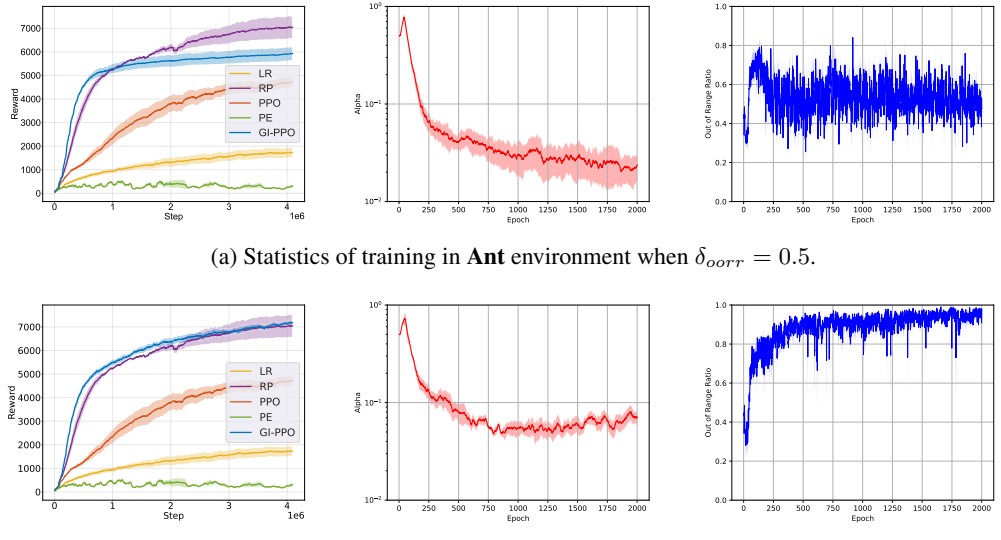

(a) Statistics of training in **Ant** environment when $\delta_{oorr} = 0.5$.

(b) Statistics of training in **Ant** environment when $\delta_{oorr} = 1.0$.

Figure 7: Change of $\alpha$ during training in Ant environment in Section 5.2. In this environment, maximum out-of-range-ratio ($\delta_{oorr}$) plays a major role in bounding $\alpha$ from growing. Observe that GI-PPO achieves better results than RP when we set $\delta_{oorr} = 1.0$.

soon decreases due to the variance and bias criteria. Compare these graphs with optimization curve in Figure 2d. Then we would be able to observe that this large $\alpha$ contributes to the faster convergence of GI-PPO compared to PPO at the early stage. However, after that, $\alpha$ decreases slowly due to the out-of-range-ratio. In the end, $\alpha$ reaches approximately $10^{-6}$, which is much lower than $10^{-3}$ of De Jong's function. These observations align well with our intuition that RP gradient would play bigger role for De Jong's function than Ackley's function, because De Jong's function exhibits RP gradients with much lower variance than those of Ackley's.

### 7.3.4 Example: Physics Simulation Problems (Section 5.2)

In Section 5.2, GI-PPO could not achieve better results than RP in Ant and Hopper environments. For Ant environment, we found out that the hyperparameter $\delta_{oorr}$ was a bottleneck that bounds GI-PPO's performance. In Figure 7, we illustrated the change of $\alpha$ during training along the change of out-of-range-ratio. When we set $\delta_{oorr} = 0.5$, $\alpha$ keeps decreasing over time to keep the ratio around $0.5$, which means the diminishing role of RP gradients in training. In contrast, when we set $\delta_{oorr} = 1.0$, after 750 epochs, $\alpha$ does not decrease anymore, and out-of-range ratio stays near 1. This means that PPO rarely affects the training – policy update is almost driven by RP gradients.

Likewise, there are environments where RP gradients are much more effective than the other policy gradients. In those situations, it would be more desirable to solely depend on RP gradients without considering PPO. Since our method considers RP gradients in the PPO viewpoint, it is hard to detect such cases yet.

### 7.4 Experimental Details

### 7.4.1 Baseline Methods

Here we explain implementation details of the baseline methods that were used for comparisons in Section 5. First, we'd like to point out that we used relatively a short time horizon to collect experience, and used the critic network for bootstrapping, instead of collecting experience for the whole trajectory for all the methods. Therefore, the collected experience could be biased. However, we chose to implement in this way, because it allows faster learning than collecting the whole trajectory in terms of number of simulation steps.

**LR** We can estimate original LR gradient as follows, using log-derivative trick [Williams and Peng, 1989, Glynn, 1990]

$$\frac{\partial \eta(\pi_\theta)}{\partial \theta} = \frac{\partial}{\partial \theta} \mathbb{E}_{s_0,a_0,...\sim\pi_\theta} \Big[ L(s_0,a_0,...) \sum_{t=0}^{\infty} \log \pi_\theta(s_0,a_0) \Big]$$

$$= \frac{\partial}{\partial \theta} \mathbb{E}_{s_0,a_0,...\sim\pi_\theta} \Big[ \sum_{t=0}^{\infty} L(s_0,a_0,...) \log \pi_\theta(s_0,a_0) \Big],$$

where $L(s_0,a_0,...) = \sum_{t=0}^{\infty} \gamma^t r(s_t,a_t)$ is the discounted accumulated return. However, this LR gradient often suffers from high variance, because $L$ varies a lot from trajectory to trajectory. To reduce such variance and faithfully compare our method to LR gradient-based method, we decided to use advantage function in the place of $L$, and particularly used GAE [Schulman et al., 2015b] as the advantage function as we used for PPO. After calculating this LR gradient, we take a gradient ascent step with pre-defined learning rate.

**RP** By using RP gradient in Appendix 7.2.2, we can perform gradient ascent as we did in LR. Also, because we use short time horizon and critic network, this method is very similar to SHAC [Xu et al., 2022]. Please refer to the paper for more details.

**PPO** Our PPO implementation is based on that of RL Games [Makoviichuk and Makoviychuk, 2022]. However, to be as fair as possible, we used the same critic network as the other methods, instead of using that implemented in the RL library. Also, we omitted several other additional losses that are not vital to PPO's formulation, such as entropy loss.

**LR+RP** After we compute LR ($\nabla_{LR}$) and RP ($\nabla_{RP}$) gradient as shown above, we can interpolate them using their sample variance as follows [Parmas et al., 2018].

$$\nabla_{LR+RP} = \nabla_{LR} \cdot \kappa_{LR} + \nabla_{RP} \cdot (1 - \kappa_{LR}),$$

$$\kappa_{LR} = \frac{\sigma_{RP}^2}{\sigma_{RP}^2 + \sigma_{LR}^2},$$

where $\sigma_{RP}$ and $\sigma_{LR}$ are sample standard deviation of RP and LR gradients. We gain these terms by computing trace of covariance matrix of the sample gradients from different trajectories.

Since we have to compute this sample statistics, we have to do multiple different backpropagations for different trajectories, which incur a lot of computation time. Also, we found out that computing covariance matrix is also time consuming when controller has a large number of parameters. Therefore, we decided to use only limited number of sample gradients (16) to compute sample variance, and also truncate the gradient to smaller length (512) to facilitate computation.

**PE** We tried to faithfully re-implement policy enhancement scheme of [Qiao et al., 2021].

### 7.4.2 Network architecture and Hyperparameters

In this section, we provide network architectures and hyperparameters that we used for experiments in Section 5. For each of the experiments, we used the same network architectures, the same length of time horizons before policy update, and the same optimization procedure for critic updates, etc. We present these common settings first for each of the problems.

For GI-PPO, there are hyperparameters for update towards $\alpha$-policy, and those for PPO update. We denote the hyperparameters for $\alpha$-policy update by appending ($\alpha$), and those for PPO update by appending (PPO). For the definition of hyperparameters for $\alpha$-policy update, please see Algorithm 1 for details.

**Function Optimization Problems (Section 5.1)** For these problems, common settings are as follows.

- Actor Network: MLP with [32, 32] layers and ELU activation function

- Critic Network: MLP with $[32, 32]$ layers and ELU activation function
- Critic Hyperparameters: Learning rate = $10^{-3}$, Iterations = 16, Batch Size = 4
- Number of parallel environments: 64
- Horizon Length: 1
- $\gamma$ (Discount factor): 0.99
- $\tau$ (GAE): 0.95

Hyperparameters for **LR** are as follows.

- Learning Rate: Dejong(1), Dejong(64) = $10^{-3}$, Ackley(1) = $10^{-4}$, Ackley(64) = $3 \cdot 10^{-4}$
- Learning Rate Scheduler: Linear[1]

Hyperparameters for **RP** are as follows.

- Learning Rate: Dejong(1), Dejong(64) = $10^{-2}$, Ackley(1), Ackley(64) = $10^{-3}$
- Learning Rate Scheduler: Linear

Hyperparameters for **LR+RP** are as follows.

- Learning Rate: Dejong(1), Dejong(64) = $10^{-3}$, Ackley(1) = $10^{-4}$, Ackley(64) = $3 \cdot 10^{-4}$
- Learning Rate Scheduler: Linear

Hyperparameters for **PPO** are as follows.

- Learning Rate: Dejong(1), Ackley(1) = $10^{-4}$, Dejong(64), Ackley(64) = $10^{-2}$
- Learning Rate Scheduler: Constant
- Batch Size for Actor Update: 64
- Number of Epochs for Actor Update: 5
- $\epsilon_{clip}$: 0.2

Hyperparameters for **GI-PPO** are as follows.

- ($\alpha$) Learning Rate: $10^{-3}$
- ($\alpha$) Batch Size for Actor Update: 64
- ($\alpha$) Number of Epochs for Actor Update: 16
- ($\alpha$) $\alpha_0$: $10^{-5}$
- ($\alpha$) $\max(\alpha)$: 1.0
- ($\alpha$) $\beta$: 1.1
- ($\alpha$) $\delta_{det}$: 0.4
- ($\alpha$) $\delta_{oorr}$: 0.5
- (PPO) Learning Rate: Dejong(1), Ackley(1) = $10^{-4}$, Dejong(64), Ackley(64) = $10^{-2}$
- (PPO) Learning Rate Scheduler: Constant
- (PPO) Batch Size for Actor Update: 64
- (PPO) Number of Epochs for Actor Update: 5
- (PPO) $\epsilon_{clip}$: 0.2

---

[1]Learning rate decreases linearly to the minimum value as learning progresses.

**Differentiable Physics Problems (Section 5.2)**   For these problems, common settings are as follows.

- Actor Network: MLP with $[64, 64]$ (Cartpole), and $[128, 64, 32]$ (Ant, Hopper) layers and ELU activation function
- Critic Network: MLP with $[64, 64]$ layers and ELU activation function
- Critic Hyperparameters: Learning rate = $10^{-3}$(Cartpole), $2 \cdot 10^{-3}$(Ant), and $2 \cdot 10^{-4}$(Hopper), Iterations = 16, Batch Size = 4
- Number of parallel environments: 64(Cartpole, Ant), 256(Hopper)
- Horizon Length: 32
- $\gamma$ (Discount factor): 0.99
- $\tau$ (GAE): 0.95

Hyperparameters for **LR** are as follows.

- Learning Rate: $10^{-4}$
- Learning Rate Scheduler: Linear

Hyperparameters for **RP** are as follows.

- Learning Rate: Cartpole = $10^{-2}$, Ant, Hopper = $2 \cdot 10^{-3}$
- Learning Rate Scheduler: Linear

Hyperparameters for **PPO** are as follows.

- Learning Rate: Cartpole = $3 \cdot 10^{-4}$, Ant, Hopper = $10^{-4}$
- Learning Rate Scheduler: Cartpole = Adaptive[2], Ant, Hopper = Constant
- Batch Size for Actor Update: 2048
- Number of Epochs for Actor Update: 5
- $\epsilon_{clip}$: 0.2

Hyperparameters for **GI-PPO** are as follows.

- ($\alpha$) Learning Rate: Cartpole = $10^{-2}$, Ant = $5 \cdot 10^{-4}$, Hopper = $5 \cdot 10^{-3}$
- ($\alpha$) Batch Size for Actor Update: Cartpole, Ant = 2048, Hopper = 8192
- ($\alpha$) Number of Epochs for Actor Update: 16
- ($\alpha$) $\alpha_0$: Cartpole, Ant = $5 \cdot 10^{-1}$, Hopper = $5 \cdot 10^{-3}$
- ($\alpha$) $\max(\alpha)$: 1.0
- ($\alpha$) $\beta$: 1.02
- ($\alpha$) $\delta_{det}$: 0.4
- ($\alpha$) $\delta_{oorr}$: Cartpole, Hopper = 0.75, Ant = 0.5
- (PPO) Learning Rate: Cartpole = $3 \cdot 10^{-4}$, Ant, Hopper = $10^{-4}$
- (PPO) Learning Rate Scheduler: Cartpole = Adaptive, Ant, Hopper = Constant
- (PPO) Batch Size for Actor Update: 2048
- (PPO) Number of Epochs for Actor Update: 5
- (PPO) $\epsilon_{clip}$: 0.2

---

[2]Learning rate is adaptively controlled, so that the KL divergence between the updated policy and the original policy is maintained at certain value, 0.008 in this case.

**Traffic Problems (Section 5.3)**   For these problems, common settings are as follows.

- Actor Network: MLP with $[512, 64, 64]$ layers and ELU activation function
- Critic Network: MLP with $[64, 64]$ layers and ELU activation function
- Critic Hyperparameters: Learning rate = $10^{-3}$, Iterations = 16, Batch Size = 4
- Number of parallel environments: 64
- Horizon Length: 32
- $\gamma$ (Discount factor): 0.99
- $\tau$ (GAE): 0.95

Hyperparameters for **LR** are as follows.

- Learning Rate: $3 \cdot 10^{-4}$
- Learning Rate Scheduler: Linear

Hyperparameters for **RP** are as follows.

- Learning Rate: $10^{-3}$
- Learning Rate Scheduler: Linear

Hyperparameters for **LR+RP** are as follows.

- Learning Rate: $3 \cdot 10^{-4}$
- Learning Rate Scheduler: Linear

Hyperparameters for **PPO** are as follows.

- Learning Rate: $3 \cdot 10^{-4}$
- Learning Rate Scheduler: Constant
- Batch Size for Actor Update: 2048
- Number of Epochs for Actor Update: 5
- $\epsilon_{clip}$: 0.2

Hyperparameters for **GI-PPO** are as follows.

- ($\alpha$) Learning Rate: $10^{-5}$
- ($\alpha$) Batch Size for Actor Update: 2048
- ($\alpha$) Number of Epochs for Actor Update: 16
- ($\alpha$) $\alpha_0$: $10^{-1}$
- ($\alpha$) $\max(\alpha)$: 1.0
- ($\alpha$) $\beta$: 1.1
- ($\alpha$) $\delta_{det}$: 0.4
- ($\alpha$) $\delta_{oorr}$: 0.5
- (PPO) Learning Rate: $3 \cdot 10^{-4}$
- (PPO) Learning Rate Scheduler: Constant
- (PPO) Batch Size for Actor Update: 2048
- (PPO) Number of Epochs for Actor Update: 5
- (PPO) $\epsilon_{clip}$: 0.2

## 7.5   Problem Definitions

Here we present details about the problems we suggested in Section 5.

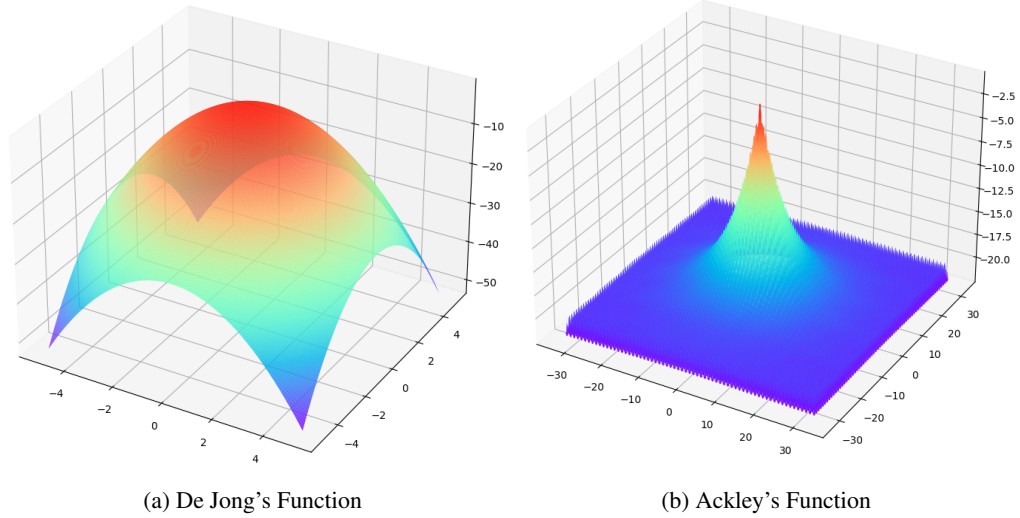

| (a) De Jong's Function | (b) Ackley's Function |

Figure 8: Landscape of target functions in 2 dimensions.

### 7.5.1 Function Optimization Problems (Section 5.1)

(N-dimensional) De Jong's function ($F_D$) and Ackley's function ($F_A$) are defined for $n$-dimensional vector $x$ and are formulated as follows [Molga and Smutnicki, 2005]:

$$F_D(x) = \sum_{i=1}^{n} x_i{}^2,$$

$$F_A(x) = -20 \cdot exp(-0.2 \cdot \sqrt{\frac{1}{n} \sum_{i=1}^{n} x_i{}^2}) -$$

$$exp(\frac{1}{n} \sum_{i=1}^{n} cos(2\pi x_i)) + 20 + exp(1).$$

As we mentioned in Section 5.1, we multiply -1 to these functions to make these problems maximization problems. Also, even though these two functions have their optimum at $x = 0$, they exhibit very different landscape as shown in Figure 8.

When it comes to the formal definition as RL environments, we can define these problems as follows.

- Episode Length: 1
- Observation: $[0]$
- Action: $n$-dimensional vector **x**, all of which element is in $[-1, 1]$.
- Reward: De Jong's Function = $F_D(5.12\mathbf{x})$, Ackley's Function = $F_A(32.768\mathbf{x})$

### 7.5.2 Traffic Problems (Section 5.3)

Before introducing pace car problem, which we specifically discussed in this paper, we'd like to briefly point out the traffic model that we used to simulate motions of individual vehicles in the traffic environment.

**Traffic Model**   In our traffic simulation, the state $s \in \mathbb{R}^{2N}$ is defined as a concatenation of all vehicle states, $q_n \in \mathbb{R}^2$, where $1 \le n \le N$ stands for vehicle ID. $q_n$ can be represented simply with the vehicle's position ($x_n$) and velocity ($v_n$).

$$q_n = \begin{bmatrix} x_n \\ v_n \end{bmatrix}$$

$$s = \begin{bmatrix} q_1^T, \ldots, q_N^T \end{bmatrix}^T,$$

where $n$ is the vehicle ID. Since this traffic state $s$ changes over time, we represent the traffic state at timestep $t$ as $s_t$. Then our differentiable traffic simulator is capable of providing the gradient of the simulation state at the next timestep $t + 1$ with respect to the state at the current timestep $t$, or $\frac{ds_{t+1}}{ds_t}$.

This is possible because we use the Intelligent Driver Model (IDM) to model car-following behavior in our simulator [Treiber et al., 2000], which is differentiable. IDM describes vehicle speed $\dot{x}$ and acceleration $\dot{v}$ as a function of desired velocity $v_0$, safe time headway in meters $T$, maximum acceleration $a$, comfortable deceleration $b$, the minimum distance between vehicles in meters $\delta$, vehicle length $l$, and difference in speed with the vehicle in front $\Delta v_\alpha$ as follows:

$$\dot{x}_\alpha = \frac{\mathrm{d}x_\alpha}{\mathrm{d}t} = v_\alpha,$$

$$\dot{v}_\alpha = \frac{\mathrm{d}v_\alpha}{\mathrm{d}t} = a \left( 1 - \left( \frac{v_\alpha}{v_0} \right)^\delta - \left( \frac{s^*(v_\alpha, \Delta v_\alpha)}{s_\alpha} \right)^2 \right),$$

$$s_\alpha = x_{\alpha-1} - x_\alpha - l_{\alpha-1},$$

$$s^*(v_\alpha, \Delta v_\alpha) = s_0 + v_\alpha T + \frac{v_\alpha \Delta v_\alpha}{2 \sqrt{a\,b}},$$

$$\Delta v_\alpha = v_\alpha - v_{\alpha-1},$$

where $\alpha$ means the order of the vehicle in the lane. Therefore, $(\alpha - 1)$-th vehicle runs right in front of $\alpha$-th vehicle, and this relationship plays an important role in IDM. Also note that the computed acceleration term ($\dot{v}$) is differentiable with respect to the other IDM variables.

IDM is just one of many car-following models used in traffic simulation literature. We choose IDM for its prevalence in previous mixed autonomy literature, however, any ODE-based car-following model will also work in our simulator as far as it is differentiable. In our simulator, automatic differentiation governs the gradient computation.

**Lane**   Under IDM, lane membership is one of the most vital variables for simulation. This is because IDM builds on the leader-follower relationship. When a vehicle changes its lane with lateral movement, as shown as red arrows in Figure 9, such relationships could change, and our gradients do not convey any information about it, because lane membership is a discrete variable in nature. Note that this lateral movement also affects the longitudinal movement, rendered in green arrows in Figure 9, of a vehicle, and gradient from it is valid as far as the lane membership does not change. However, when the lane membership changes, even if our simulator gives gradient, it tells us only "partial" information in the sense that it only gives information about longitudinal behavior. Therefore, we could say that analytical gradients we get from this environment is "incomplete" (not useless at all, because it still gives us information about longitudinal movement), and thus biased.

**Pace Car Problem**   In this problem, there is a single autonomous vehicle that we have to control to regulate other human driven vehicles, which follow IDM, to run at the given target speed. Since human driven vehicles control their speeds based on the relationship to their leading vehicles, our autonomous vehicle has to control itself to run in front of those human driven vehicles and adjust their speeds to the target speed. The number of lanes and human driven vehicles varies across different environments as follows.

- Single Lane: Number of lanes = 1, Number of vehicles per lane = 1
- 2 Lanes: Number of lanes = 2, Number of vehicles per lane = 2
- 4 Lanes: Number of lanes = 4, Number of vehicles per lane = 4
- 10 Lanes: Number of lanes = 10, Number of vehicles per lane = 1

See Figure 10 for 10-lane environment.

Then we can formally define RL environments as follows, where $N$ is the number of human driven vehicles in total, and $v_{tgt}$ is the target speed.

- Episode Length: 1000
- Observation: $s \in \mathbb{R}^{2N}$
- Action: $\mathbf{a} \in \mathbb{R}^2$ ($\mathbf{a}_0$ = Acceleration, $\mathbf{a}_1$ = Steering)
- Reward: $1 - \frac{1}{N} \sum_{i=1}^{N} \min(\frac{|v_i - v_{tgt}|}{v_{tgt}}, 1)$

However, to facilitate learning, we additionally adopted following termination conditions to finish unpromising trials early.

- Terminate when autonomous vehicle collides with human driven vehicle
- Terminate when autonomous vehicle goes out of lane
- Terminate when autonomous vehicle runs too far away from the other vehicles
- Terminate when autonomous vehicle falls behind the other vehicles

When one of these conditions is met, it gets reward of $-1$ and the trial terminates.

## 7.6 Renderings

### 7.6.1 Traffic Simulation

**Gradient flow**    As described in Appendix 7.5.2, analytical gradients are only valid along longitudinal direction, as illustrated with green arrows in Figure 9. When a vehicle changes lane with lateral movement, as shown with red arrows, even though such change would incur different behaviors of following vehicles in multiple lanes, analytical gradients cannot convey such information. However, they still tell us information about longitudinal behavior - which is one of the motivations of our research, to use analytical gradients even in these biased environments.

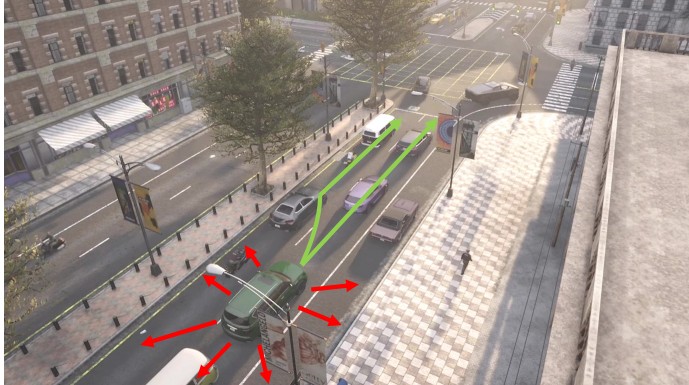

Figure 9: **Traffic Environment.** In traffic flows, only partial gradients of forward simulation are known; acceleration along a (longitudinal) traffic lane is continuous and admits gradient flow (green), while (lateral) lane-changing is a discrete event and thus prohibits gradient flow (red). Analytical gradients convey only partial information about traffic dynamics.

**Pace Car Problem**    We additionally provide renderings of the 10-lane pace car problem defined in Appendix 7.5.2. In Figure 10, we can observe 10 parallel lanes, 10 human driven vehicles rendered in yellow, and the autonomous vehicle rendered in blue that we have to control. As shown there, human driven vehicles adjust their speeds based on IDM when the autonomous vehicle blocks their way. Therefore, the autonomous vehicle has to learn how to change their lanes, and also how to run in appropriate speed to regulate the following vehicles. Our experimental results in Section 5.3 show that our method achieves better score than the baseline methods by adopting biased analytical gradients in PPO.

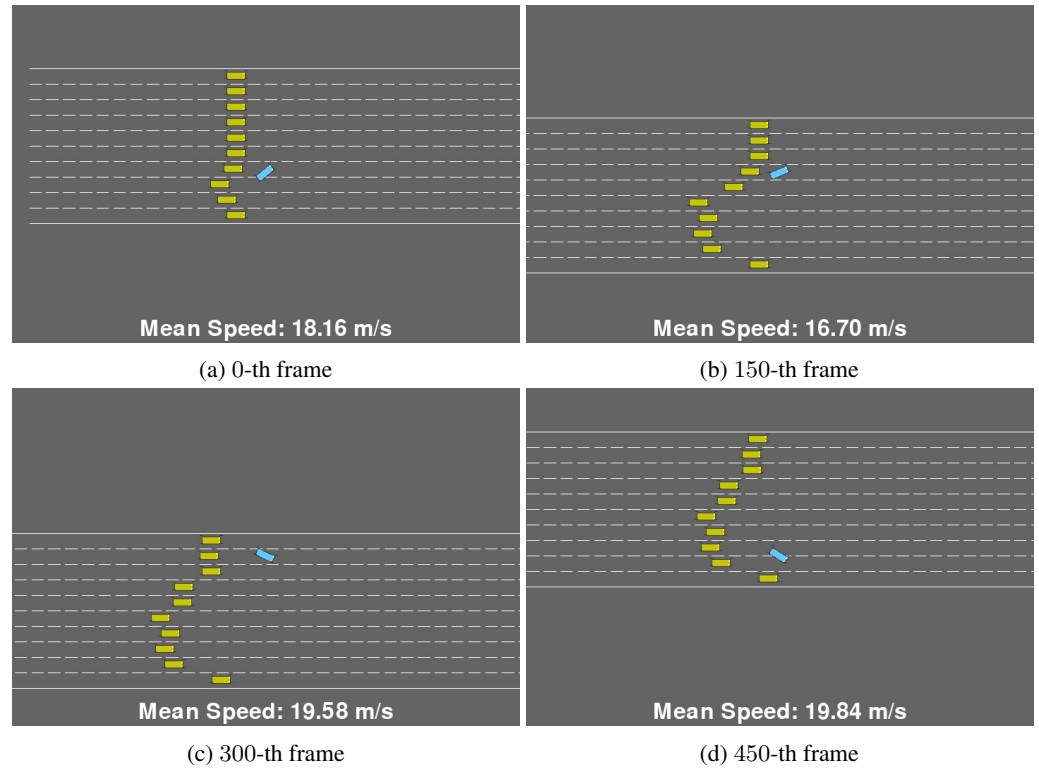

| | |
|:---:|:---:|
| Mean Speed: 18.16 m/s | Mean Speed: 16.70 m/s |
| (a) 0-th frame | (b) 150-th frame |
| Mean Speed: 19.58 m/s | Mean Speed: 19.84 m/s |
| (c) 300-th frame | (d) 450-th frame |

Figure 10: 10-Lane pace car environment. The autonomous vehicle (Blue) has to interfere paths of the other human driven vehicles (Yellow) to limit their speeds to $10m/s$. Even though it is a hard problem to achieve high score, our method achieved far better score than the baseline methods.

## 7.7 Computational Costs

We provide wall clock training times for each method in traffic environments to finish all the training epochs in Table 2. All of the trainings were done based on the settings in Appendix 7.5.2. Note that the longer training time does not always mean that the method is worse than the others – the method could have achieved better results in shorter time, while requiring much longer time to complete all the training epochs.

In Table 2, we can observe that GI-PPO requires a little more time than RP, as it computes analytical gradients as RP does, but requires an additional time for controlling $\alpha$ and doing PPO update. However, the additional cost is not so significant. In contrast, LR+RP consumes much more time than the other methods, even though it is based on the same spirit as ours. This is because it has to backpropagate through each trajectory in the experience buffer to estimate the sample variance of the gradients. Therefore, we can see that our method attains better results than LR+RP (Figure 4) with lower computational cost.

Table 2: Average (wall clock) training time (sec) for traffic problems.

| Problem | LR | RP | PPO | LR+RP | GI-PPO |
|---|---|---|---|---|---|
| Single Lane | 120 | 282 | 181 | 1335 | 332 |
| 2 Lanes | 142 | 330 | 218 | 1513 | 411 |
| 4 Lanes | 304 | 646 | 366 | 2093 | 813 |
| 10 Lanes | 244 | 496 | 294 | 2103 | 533 |

