# OpenReview forum: "Gradient Informed Proximal Policy Optimization"
_NeurIPS.cc/2023/Conference — NeurIPS 2023 poster_

### Official Review · Reviewer_39SH · 2023-06-26

**Soundness:** 3 good
**Presentation:** 3 good
**Contribution:** 2 fair
**Rating:** 5
**Confidence:** 3

**Summary:**

This paper studied the combined use of both the analytical policy gradient and the likelihood ratio policy gradient for training policy networks, based on the PPO algorithm. To make the combination feasible, a new alpha-policy is introduced and its approximation technique has been successfully developed. Besides some theoretical studies, empirical results further show the potential usefulness of the newly proposed algorithm in solving some benchmark problems.

**Strengths:**

The theoretical study in this paper regarding the analytical policy gradient and the alpha-policy sounds interesting and novel. The experiment results show that the new algorithm can be very useful on some benchmark problems.

**Weaknesses:**

While it is interesting to mix analytical policy gradients with learned policy gradients to enhance the reliability and performance of the policy network training algorithms, this idea is clearly not restricted to on-policy algorithms such as PPO. Although the authors made it clear that PPO is one of the most popularly used on-policy algorithms, it remains questionable why this paper only studies the effectiveness of using combined policy gradients in PPO. The possibility and potential limitations of using the proposed policy gradient combination technique on other algorithms, particularly off-policy algorithms, may need to be further justified and investigated.

This paper requires prior knowledge of the environment dynamics that must be differentiable (or partially differentiable) in nature. Many real-world reinforcement learning problems may not satisfy this requirement. Hence, the practical usefulness of the new algorithm remains a bit questionable. The authors are highly recommended to clearly evaluate and justify the practical value of the new algorithm.

Furthermore, if the environment model is known in advance, it is possible to conduct effective model-based policy training without using analytical policy gradients. It remains unclear to me what the advantages of using analytic policy gradient techniques would be, compared to planning and other model-based reinforcement learning techniques. It is also desirable if the performance strength of using the analytic policy gradients, compared to other model-based reinforcement learning methods, can be experimentally evaluated and reported in this paper.

The authors stated several times in the paper that the analytical policy gradients would be highly biased when they are incomplete. I don't understand what this means and why it is true. Incompleteness and bias present two separate dimensions regarding the quality of analytical policy gradients. Why should they be strongly correlated?

Some mathematical claims require more clarity. For example, Lemma 4.4 mentioned a particular advantage function A-hat. However, the actual definition of this function is not presented, making it hard to accurately understand this lemma and its applicability.

Some critical details were missing regarding the new algorithm design in the main text. For example, the new algorithm requires you to adjust alpha from time to time. However, it is not clear how alpha is actually adjusted. The formula for adjusting alpha is not presented and clearly justified in the main text. Meanwhile, some claims in the algorithm design also need strong justifications. For example, it is stated on page 6 that by constraining the determinant of a matrix to be near 1, we can guarantee stable policy updates. However, the validity of this statement is not proved or properly explained. Moreover, if all eigenvalues are far greater than 1, alpha would be set to be very close to 0. In this case, would the analytical gradient become useless? Similarly, it is not clear why the ratio of the two determinants gives us the difference between the two policies.

According to the experiment results presented in the main text, it seems that the new algorithm is only useful when the problem being solved has many poor local optima (hence the algorithm needs to inject some uncertainties or conducts more explorations) or when the complete analytical policy gradient cannot be obtained. On problems of differentiable physics simulations, using the analytical policy gradients alone appear to be able to achieve the best results. In view of this, the real practical value of the new algorithm may need to be further investigated on more benchmark problems.

**Questions:**

What are the possibilities and potential limitations of using the proposed policy gradient combination technique on other algorithms, particularly off-policy algorithms?

What is the practical impact of the assumption regarding the prior knowledge of the environment dynamics that must be differentiable?

What are the theoretical and practical advantages of the new algorithm design, compared to other model-based reinforcement learning methods?

Incompleteness and bias present two separate dimensions regarding the quality of analytical policy gradients. Why should they be strongly correlated?

**Limitations:**

I do not have any concerns regarding this question.

---

> ### Author Rebuttal · Authors · 2023-08-09
>
> Thank you for your valuable comments, we really appreciate them.
>
> **Q1. What are the possibilities and potential limitations of using the proposed policy gradient combination technique on other algorithms, particularly off-policy algorithms?**
>
> A1. We'd like to emphasize that **on-policy algorithms operate on a very different theoretical foundation from off-policy learning algorithms**. While on-policy algorithms estimate policy gradients that are only valid for the current policy, off-policy algorithms do not use them. Therefore, incorporating analytical gradients into off-policy algorithms requires a shift in theoretical perspective from ours. Moreover, there are already several works to which we can refer when it comes to off-policy algorithms [1]. While the idea of using analytical gradients to enhance off-policy learning algorithms is compelling, it is beyond the scope of this work and deserves a separate investigation.
>
> **Q2. What is the practical impact of the assumption regarding the prior knowledge of the environment dynamics that must be differentiable?**
>
> A2. Please refer to A4 of global rebuttal at the top.
>
> **Q3. What are the theoretical and practical advantages of the new algorithm design, compared to other model-based reinforcement learning methods?**
>
> A3. To the best of our knowledge, model-based RL methods approximate world dynamics from sample trajectories, which require another optimization procedure. Because of this, [2] reported that the SE-MBPO, which is one of the model-based RL methods augmented with analytical gradients, requires a lot more wall-clock time for training (8 hour) than the RP method (15 min) like ours in Ant environment. Therefore, our method is expected to be more efficient than other model-based RL methods.
>
> **Q4. Incompleteness and bias present two separate dimensions regarding the quality of analytical policy gradients. Why should they be strongly correlated?**
>
> A4. About the relationship between the two terms, we believe that "bias" is a broader term that includes "incompleteness". "Bias" in this context means that we cannot estimate the correct gradient even if we have infinite number of samples. In the traffic environments that we discussed, **gradients convey only partial information about world dynamics**. To be specific, they convey information along the lanes, but not across lanes. Please refer to Appendix 7.5.2. This is the reason why we called the gradients "incomplete", and under this circumstance we cannot estimate correct gradients even if there are infinite samples, which dictates their biases.
>
> **Q5. Lemma 4.4 mentioned a particular advantage function $\hat{A}$, but its actual definition is not presented.**
>
> A5. The definition of $\hat{A}$ is presented in Appendix 7.1.5. Please refer to it for details.
>
> **Q6. It is not clear how alpha is actually adjusted.**
>
> A6. There is a full pseudocode for our algorithm in Appendix 7.3.3, which illustrates how alpha is adjusted during training in detail. Please let us know if it is still unclear.
>
> **Q7. The claim that we can guarantee stable policy updates by constraining the determinant of a matrix to be near 1 is not proved or properly explained.**
>
> A7. Thank you for the comment, but we'd like to say that it is **one of necessary conditions that we have to satisfy to make policy updates more stable**, and we already suggested three reasonings for this claim. To briefly reiterate them, we proved that the determinant should stay near 1 to make $\alpha$ policy not become an invalid policy. Also, as the determinant is related to variance of the RP gradients, (Figure 2), we have to keep it close to 1 to constrain the variance of analytical gradients used in policy updates. Finally, it helps us keep $\alpha$ policy near the current policy - please refer to A9.
>
> **Q8. If all eigenvalues are far greater than 1, $\alpha$ would be set to be very close to 0. In this case, would the analytical gradient become useless?**
>
> A8. Yes, this is our intention. If all eigenvalues are far greater than 1, it means that the variance of the analytical gradients  is very large, and thus it's very likely that the policy updates based on them would be unstable. In such cases, we decrease alpha to nearly 0, and rely mostly on PPO. This is the same spirit as others do when mixing LR and RP gradients for policy updates - if RP gradients have far higher variance than LR, they can be neglected [3].
>
> **Q9. It is not clear why the ratio of the two determinants gives us the difference between the two policies.**
>
> A9. When $\alpha = 0$, it is trivial that the ratio equals to 1. Therefore, we can say that the deviation of the ratio from 1 is one of (indirect) sufficient conditions for detecting differences between the two policies, rather than necessary conditions.
>
> **Q10. It seems that the new algorithm is only useful when the problem being solved has many poor local optima or when the complete analytical policy gradient cannot be obtained.**
>
> A10.  In fact, our research is motivated by the observation that many environments we counter have these very properties you mentioned, thus this work is widely applicable. For instance, for rigid body physics simulation, there are a lot of discontinuities incurred due to collisions, which require special techniques to make them differentiable. Instead of doing that, we design a policy learning algorithm that can leverage (possibly biased) analytical gradients, because it is quite time-consuming to make every environment fully differentiable. We expect our algorithm could become a good baseline algorithm to test with only partial differentiability, which is much easier to obtain than full differentiability.
>
> [1] Qiao, Yi-Ling, et al. "Efficient differentiable simulation of articulated bodies."
>
> [2] Xu, Jie, et al. "Accelerated policy learning with parallel differentiable simulation."
>
> [3] Parmas, Paavo, et al. "PIPPS: Flexible model-based policy search robust to the curse of chaos."

---

> ### Comment · Reviewer_39SH · 2023-08-11
> **Thank the authors for their response**
>
> I would like to thank the authors for their response, which has addressed some of my concerns. I will increase my rating a bit. In the meantime, I am not fully convinced by the practical applicability of the algorithm. Regarding the discussion on model-based RL, my point is when the model is known (hence no cost is involved in learning the model), similar to the condition of the newly proposed algorithm, how well the new algorithm can outperform model-based RL in terms of both theoretical and empirical advantages. Additionally, I still don't understand the difference between completeness and bias based on the authors' explanation. I think a more thorough mathematical definition is required. This is also the case regarding the explanation on the ratio of the two determinants and why this can measure the difference between two policies. Finally, I believe the main text of the paper should be self-contained, without relying on any appendices. Hence, the definition of A hat should be in the main text.

---

> > ### Author Response · Authors · 2023-08-14
> >
> > Thank you for raising the rating and additional comments, we appreciate them! Here are our answers to additional questions.
> >
> > **Q1. When the model is known, hence no cost is involved in learning the model, how well the new algorithm can outperform model-based RL in terms of both theoretical and empirical advantages.**
> >
> > A1. According to [1], we believe that your suggested model-based RL with known models falls into the same category as models like AlphaZero [2], which leverage planning methods such as Monte Carlo Tree Search (MCTS). To briefly recap how MCTS works: since the world dynamics model (which are the Go rules for AlphaZero) is fully known, it "simulates" future steps of gameplay using the estimations of its neural network as a prior. That is, it rolls out future gameplay steps based on rough suggestions from the neural networks, builds a search tree based on them, and selects the action that yields the best reward. In this approach, the model is fully known and is thus used as a tool to simulate various scenarios.
> >
> > However, we’d like to note that this strategy works in a very different environment than ours - it works in the **discrete** action space, while our method is for **continuous** one. It is unclear if we can apply planning methods like MCTS to our case or not, and thus it is difficult to compare it directly to our method. In spite of that, we’d like to point out that even if we could apply MCTS-like planning methods to our physics simulation problems, it could require much more computational cost for training. This is because it has to expand its search tree by taking a large number of timesteps, and take a SINGLE action based on it. For instance, AlphaZero runs 1600 simulation steps to build a search tree for determining a SINGLE action. Considering that the most of the computational cost comes from the expensive simulation costs for physics simulations, it is prohibitive to run such a number of simulation steps for determining action at a single time step, and thus training the network. Therefore, we’d like to underline the difficulty to directly compare model-based RL with a fully known model to our method.  At the same time, the planning method like MCTS used for such model-based RL methods could require much more computational cost than ours in the worst case scenario.
> >
> > Exploring how analytical gradients can be efficiently applied to model-based RL methods would be another very exciting research direction.
> >
> > **Q2. I still don't understand the difference between completeness and bias based on the authors' explanation. I think a more thorough mathematical definition is required. This is also the case regarding the explanation on the ratio of the two determinants and why this can measure the difference between two policies.**
> >
> > A2. First, after some reconsideration, we realize that ‘completeness' is not an appropriate term to use in the broader, general context. We will remove the term in the revision and explain our method using bias only. However, it does not change our main claim that our method can leverage possibly biased analytical gradients better than the other baseline methods. Thank you again for pointing it out to improve the exposition of the paper.
> >
> > About the second issue, we’d like to reemphasize that the ratio of the two determinants does not “measure” the difference between the two policies. Instead, our view is that it is more likely that the two policies are more different from one another when the ratio deviates more from 1, at least locally, because the ratio is 1 when the two policies are the same. In the revision, empirical results will be included to support this claim.
> >
> > **Q3. I believe the main text of the paper should be self-contained, without relying on any appendices. Hence, the definition of A hat should be in the main text.**
> >
> > A3. We agree with your point, and will move the definition of A hat to the main paper. Thank you for your suggestions.
> >
> > [1] Moerland, Thomas M., et al. "Model-based reinforcement learning: A survey."
> >
> > [2] Silver, David, et al. "Mastering the game of go without human knowledge."

---

> > > ### Comment · Reviewer_39SH · 2023-08-18
> > > **Thank the authors for the further clarification**
> > >
> > > I would like to thank the authors for the further clarification. Regarding model based RL, I am actually referring to existing works such as below
> > >
> > > M. Janner, J. Fu, M. Zhang, and S. Levine. When to trust your model: Model-based policy optimization. Advances in neural information processing systems, 32, 2019.
> > >
> > > Assuming that the model is known in advance without uncertainty, I think a direct comparison with such model-based policy optimization algorithms is possible.

---

> > > > ### Author Response · Authors · 2023-08-18
> > > >
> > > > Thank you for the reviewer's comment. First, we’d like to highlight the fact that MBPO algorithm in the suggested paper leverages (an ensemble of) neural networks to **approximate the environment dynamics**. Therefore, **it does not assume the fully known model**. In fact, if you look into the A3 of the first rebuttal above, we referred to SE-MBPO, which is an extended version of MBPO using analytical gradients. Please refer to [1] for more information. Therefore, it is not a proper method that we can use when we assume the fully known model. Even if we might be able to use the fully known model in the place of approximated model in the algorithm, then we expect that the algorithm would not be different from Soft-Actor-Critic (SAC) algorithm, which MBPO uses for backbone algorithm for policy learning.
> > > >
> > > > Also, we'd like to emphasize that our method does not assume we have a fully known model at hand. If we can get analytical gradients in another way, that would also work - for instance, we can approximate the world dynamics model with neural networks as MBPO does, and leverage analytical gradients coming from the approximated model. However, we believe that it is a quite different setting than ours, because now we have to consider the model approximation, which introduces a lot more things to consider in the training process. Along with leveraging Monte Carlo Tree Search (MCTS), these are all exciting future directions to further explore the trade-off between accuracy and efficiency, possibly robustness and generalization, in exploiting use of analytical gradients in model-based RL.
> > > >
> > > > [1] Qiao, Yi-Ling, et al. "Efficient differentiable simulation of articulated bodies."

---

### Official Review · Reviewer_DV21 · 2023-07-03

**Soundness:** 3 good
**Presentation:** 3 good
**Contribution:** 3 good
**Rating:** 6
**Confidence:** 4

**Summary:**

This paper introduces a novel policy learning method that adopts analytical gradients into the PPO framework without the necessity of estimating LR gradients. The authors introduce an adaptive α-policy that allows us to dynamically adjust the influence of analytical gradients by evaluating the variance and bias of these gradients. The experiments show that the proposed method, GI-PPO, strikes a balance between analytical gradient-based policy updates and PPO-based updates, yielding solid results compared to baseline algorithms in various environments.


**Strengths:**

This paper introduces original ideas about how to incorporate analytical gradients into the PPO framework, taking their variance and bias into account. As a result, a locally superior policy, adaptive $\alpha$-policy is introduced and fits into PPO framework seamlessly, by dynamically adjusting the influence of analytical gradients. This paper is of solid technical quality. It provides theoretical derivations and analysis to demonstrate how $\alpha$-policy fits into PPO framework seamlessly, the relationship between reparameterization RP gradient and $\alpha$-policy, and theoretical guidance to adjust $\alpha$ based on variance and bias. The experiments are extensive in general, demonstrating the solid performance compared to baseline in different environments. Last, the paper is clearly written in general.




**Weaknesses:**

In Section 5.1 (page 7, line 229), Figure [8] is missing in main paper, although we may find the figure in Appendix. In Fig. 3, the plot is too small to distinguish the relative performance for different algorithms. In Section 5.2 (page 8), there is typo line 259,  it should be Figure [4] instead of Figure [5] for Differential Physics Simulation. In Fig. 4(b,c), GI-PPO is no better than RP for tasks Ant and Hopper in term of returning rewards. To strengthen the impact of the work, it might be better to add more challenging tasks, for example, Humanoid in Mujoco environment.


**Questions:**

This paper is well written in general. However, for the readers to better understand the algorithm, can we move Algorithm 1 GI-PPO from Appendix and add into main paper? To utilize analytic gradients, it seems GI-PPO needs more extra computations from back-propagation. Can the author provide the computational cost comparison to different baseline algorithms?


**Limitations:**

There is no code provided to reproduce the results.

---

> ### Author Rebuttal · Authors · 2023-08-09
>
> Thank you for your valuable comments and support. Here we address your key concerns.
>
> **Q1. In Section 5.1, Figure 8 is missing in main paper, although we may find the figure in Appendix. Also, the plots in Figure 3 are too small to distinguish the relative performance of different algorithms.**
>
> A1. Thank you for the comment.  We will rearrange the figures to better align with the manuscript text, and enlarge them as well.  The current appearance is mainly due to the page limit and latex placement.
>
> **Q2. There is a typo in line 259, it should be Figure 4 instead of Figure 5 for Differential Physics Simulation.**
>
> A2. Thank you for finding the typo! We will fix it in the revised version.
>
> **Q3. In Figure 4, GI-PPO is no better than RP for tasks Ant and Hopper in term of returning rewards.**
>
> A3. Please refer to A2 of global rebuttal at the top.
>
> **Q4. To strengthen the impact of the work, it might be better to add more challenging tasks, for example, Humanoid in Mujoco environment.**
>
> A4. Thank you for the suggestions to further improve the exposition of this paper.  We will add those results in the revised paper.
>
> **Q5. Can we move Algorithm 1 GI-PPO from Appendix and add into main paper?**
>
> A5. We can definitely do that, but for now the page limit makes it hardly possible. If we can get additional pages, we will move it into main paper from the supplementary document.
>
> **Q6. Can the author provide the computational cost comparison to different baseline algorithms?**
>
> A6. Thank you for the question, we can add computational cost comparison in supplementary material for sure.  Briefly, GI-PPO only needs additional computational cost for PPO-based update than the RP method. Considering that the most of the computational cost comes from simulation steps and backpropagation for RP, such additional cost is not much.

---

> > ### Comment · Reviewer_DV21 · 2023-08-15
> > **Thanks to the authors for the rebuttal**
> >
> > I’ve read comments from all the other reviewers. Thank you for your rebuttal, and I appreciate that my concerns have been addressed.

---

> > > ### Author Response · Authors · 2023-08-15
> > >
> > > Thank reviwer(s) for valuable comments, we are happy that we could address reviewer(s)' concerns!

---

### Official Review · Reviewer_8kSu · 2023-07-12

**Soundness:** 2 fair
**Presentation:** 1 poor
**Contribution:** 3 good
**Rating:** 4
**Confidence:** 4

**Summary:**

This paper integrates analytical gradients with PPO. They introduce an \alpha-policy to control the bias and variance during the update of the policy. Results on some differentiable environments show the effectiveness of the proposed method.

**Strengths:**

This paper study how to incorporate analytical gradient into existing Rl algorithm, which is interesting.

**Weaknesses:**

The main weakness of the paper is that the writing of the paper is somewhat poor.
The presentation lacks motivation or necessary intuitive explanations. There are some imprecise statements and typos (see below).
These flaws make the paper hard to follow.

Major concerns:
- This paper claims that it's a combination of analytical gradient and PPO. However, I didn't see where you use PPO (eq. 4 is not PPO!).
- Some statements lack motivation. I don't understand why the algorithm is composed of three parts (Line 182). Could you give more explanation? And why the adjusting $\alpha$ should be designed in that way? Besides, it would be good if there was more explanation of Def 4.1
- The proposed method does not perform well on the physical simulation tasks. In Figure 4, the proposed method performs worse than the RP method on two high-dimensional tasks: Ant and Hopper.
- In most of the practical tasks, it is hard to get the analytical gradient. This limits the potential of the proposed method. The paper only does experiments in simulation tasks. Could you give more examples of where we can get analytical gradients in practice?

There are some minor technical issues in the paper:
- Line 106: The symbol $A$ is repeated with the action space $A$ (in Line 97).
- Line 97: You assume the probability to be within $[0,1]$. This means that the action space is discrete. However, in Line 101, you assume action space is continuous.


Some statements in the paper are wrong:
- "PPO relies on the observation that we can evaluate a policy πθ with our current policy πθ¯". This is not right. Because the $\rho_{\pi_\theta}(s)$ is not available.
- Line 137: "the difference between πθ¯ and πθ must be small enough to get an accurate estimate." In fact, in the theory of TRPO, they introduce a lower bound with KL divergence penalty term. They don't care whether that's an accurate estimate.

**Questions:**

- What do you mean by inherent chaotic nature (Line 25)? Could you please give some examples?
- I'm confused about how is eq. 2 used in your method.
Please also see my comments in the weakness.

---

> ### Author Rebuttal · Authors · 2023-08-08
>
> Thank you for your comments.  We address some possible confusions below.
>
> **Q1. How is PPO used in this paper? Eq. 4 is not PPO, and description about PPO, especially that related to availability of $\rho_{\pi_{\theta}}(s)$, is wrong.**
>
> A1. Though the exact formulation is different, the **main objective function of both TRPO and PPO is same as Eq. 4**. The exact formulation differs between these two algorithms because they opt for different ways to constrain the policy optimization space. To be specific, if we see Eq. 14 in [1] (TRPO), we can observe that the objective function is the MC estimate of our Eq. 4, and it imposes KL divergence constraint to limit optimization space. For PPO, as seen Eq. 7 in [2], we note that it also optimizes MC estimate of our Eq. 4, but clips the estimates for constraining optimization space. Therefore, we'd like to suggest that Eq. 4 explains the objective function of PPO well, and thus is a good theoretical basis to show how we can incorporate analytical gradients to PPO framework.
>
> Additionally, we note that $\rho_{\pi_{\theta}}(s)$ is not available, given collected experience with $\pi_{\bar{\theta}}$. This is the **exactly the reason why we use $\rho_{\pi_{\bar{\theta}}}(s)$ instead of $\rho_{\pi_{\theta}}(s)$, as shown in Eq. 4 and Eq. 3 in [1]**. If we replace $\rho_{\pi_{\theta}}(s)$ with $\rho_{\pi_{\bar{\theta}}}(s)$, then it becomes the first order approximation of the expected return of $\pi_{\theta}$ [3]. In that sense, we stated that "we can evaluate a policy $\pi_{\theta}$ with our current policy $\pi_{\bar{\theta}}$", even if we do not have access to $\rho_{\pi_{\theta}}(s)$.
>
> **Q2. Why is the algorithm composed of three parts as shown in Line 182?**
>
> A2. We’d like to reemphasize that we use **variance and bias of analytical gradients** to control their role in policy update. It has been already used in several other literature [4], [5] as valid criteria to evaluate the validity of (analytical) RP gradients against LR gradients. However, **since we do not have LR gradients to compare, we need another measure to evaluate the variance and bias of analytical gradients**.
>
> To that end, we suggested $\alpha$ policy. Based on our propositions, **we can evaluate the variance and bias of analytical gradients if we have $\alpha$ policy**. Therefore, we first need to update our policy toward $\alpha$ policy - which is the first step of our algorithm. After approximating $\alpha$ policy, we evaluate the variance and bias of analytical gradients using criteria suggested in the paper, which is the second step. In this step, if our criteria is not met, we then decrease $\alpha$ so that analytical gradients do not contribute to the policy update much in the next iteration. Finally, we conduct policy updates based on PPO - in fact, as we have shown in the paper, this step is not necessary if the analytical gradients are correct, because $\alpha$ policy is locally better than the current policy in PPO viewpoint (Proposition 4.2). However, if the analytical gradients are flawed and thus $\alpha$ goes to 0, our policy update will be very slow without PPO. Therefore, we can **consider PPO as a safeguard, which guarantees certain amount of policy update even when the analytical gradients are not reliable**.
>
> **Q3. Why $\alpha$ is adjusted in the way suggested in the paper?**
>
> A3. Please refer to A3 of global rebuttal at the top.
>
> **Q4. It would be good if there was more explanation of Def 4.1.**
>
> A4. Def 4.1 introduces the concept of $\alpha$ policy, which can be considered as a **new policy that selects slightly better action than the old policy with the same probability (intuitively)**. We will add more explanation, and move Figure 8 in Appendix 7.6.1 to the main paper if possible to help understanding.
>
> **Q5. The proposed method does not perform well on the two physical simulation tasks.**
>
> A5. Please refer to A2 of global rebuttal at the top.
>
> **Q6. How can we apply this method to practical tasks?**
>
> A6. Please refer to A4 of global rebuttal at the top.
>
> **Q7. Minor technical issues in Line 97 and 106**
>
> A7. Thank you for pointing them out.  We will correct them in the revised version.
>
> **Q8. Line 137: Wrong description about $\pi_{\bar{\theta}}$ and $\pi_{\theta}$.**
>
> A8. TRPO [1] suggests the lower bound based on the observation that they can use $L_{\pi_{old}}(\pi_{new})$ to approximate the desired $\eta(\pi_{new})$ up to the first order. Furthermore, in their practical algorithm, they constrain the optimization space by upper bounding KL divergence, instead of using it as a penalty term. Likewise, **their logic is grounded on the fact that we can locally approximate the expected return of the new policy using the old one**, and in this sense, we'd like to respectfully suggest that our description was indeed correct.
>
> **Q9. What is inherent chaotic nature?**
>
> A9. The term in this context mainly describes exploding variance of the analytical gradients. We'd like to introduce related papers that discuss this topic in detail: [4], [6].
>
> **Q10. How Eq. 2 is used is confusing.**
>
> A10. Eq. 2 shows the basic gradients that our differentiable environments provide us. With those gradients, we can compute gradients of advantage value w.r.t. the actions ($\frac{dA}{da}$). If we use GAE for estimating advantage, we can compute the gradients using formulas provided in Appendix 7.2.1. In the formula, note that we use $\frac{d \delta}{da}$, which can be computed using the basic gradients in eq. 2, as $\delta$ is defined using $r$ and $s$.
>
> [1] Schulman, John, et al. "Trust region policy optimization."
>
> [2] Schulman, John, et al. "Proximal policy optimization algorithms."
>
> [3] Kakade, Sham M. "A natural policy gradient."
>
> [4] Parmas, Paavo, et al. "PIPPS: Flexible model-based policy search robust to the curse of chaos."
>
> [5] Suh, Hyung Ju, et al. "Do differentiable simulators give better policy gradients?."
>
> [6] Metz, Luke, et al. "Gradients are not all you need."

---

> ### Author Response · Authors · 2023-08-16
>
> Dear Reviewer 8kSu,
>
> Did our rebuttal address issues you raised?   Do you still have any more questions?  Thank you.
>
> Best,
>    The Authors

---

### Official Review · Reviewer_WNVi · 2023-07-20

**Soundness:** 3 good
**Presentation:** 4 excellent
**Contribution:** 3 good
**Rating:** 7
**Confidence:** 4

**Summary:**

In this manuscript, the author(s) combined the PPO framework with analytical gradients and proposed a policy learning method to learn better policy quickly. Through empirical experiments, the author(s) demonstrated that the proposed method was superior to baseline methods in several scenarios.




**Strengths:**

### Strong points:

1) Utilizing $\alpha$-policy to deal with the influence of analytical policy gradients when training;

2) Adopting criteria to detect/calculate variance and bias;

3) An algorithm was designed to handle the strength of analytical gradients during updating;

4) Empirical results look good.

By the way, in the Supplementary Materials, the author(s) provided a mp4 video file, where the animation on the scenario of the traffic problem looks fantastic. Thanks for providing that.




**Weaknesses:**

### Weak points:

Although the work of this manuscript looks interesting, some points in this manuscript are not so clear to me, I hope the author(s) can explain/detail them. Thanks.

1) Some descriptions of empirical experiments are not so clear.

2) Missing codes related to the proposed method in this manuscript.

3) If possible, some theoretical results in this manuscript should be empirically verified.




**Questions:**


Detailedly, I have the following comments/questions:

1) For the description or empirical experiment details, some places in this manuscript seem unclear. For example,

(1) Definition 5 was mentioned several times in the manuscript, such as, in Lines 179, 188, and even in the Supplementary Materials, Line 419, but where is Definition 5?

(2) The proof is mainly for Proposition, Lemma, Theorem, etc. In Line 419 of Supplementary Materials, it mentioned "Proof of Definition 5", I am not clear.

(3) Could you please explain/clarify the fluctuation of the blue curve (i.e., GI-PPO) in Figure 3 (c)? Why caused this phenomenon?

(4) The order number (or citation) of Figures in this manuscript is messy. For instance, in Line 229, Figure 8 is in the Supplementary Materials, right?  Figure 1 seems to be cited until Line 245?

2) If possible, some theoretical results, for example, Proposition 4.5, can be verified by empirical experiments?

3) Some other tiny issues/typos:

(1) The font size used for the coordinates in Figure is too small.

(2) The format of mathematical formulas in this manuscript is inconsistent. Some end with punctuation, and some don't.

(3) Many issues in References. For instance, Sometimes the conference name is the full name plus the abbreviation, and sometimes only the full name; Missed source of the cited references; and so on. Hope the author(s) will check carefully and correct them.

In addition, I did not scrutiny the proofs step-by-step, but I think the proofs should be Ok.

Thank the author(s) for submitting this interesting work. Except for the several possible issues/questions mentioned above. I think this is a basically good manuscript. I look forward to the response from the author(s). If the author feels any of my comments are inappropriate, please feel free to point them out :-) Thanks.





**Limitations:**

Yes. As the author(s) revealed, there might be several limitations, such as(1) How to approximate $\alpha$-policy efficiently; (2) Adjusting PPO's clipping range; (3) Computational efficiency. Hope that the author(s) can further investigate these limitations and solve these potential problems in the near future. Good luck!

---

> ### Author Rebuttal · Authors · 2023-08-08
>
> Thank you for your valuable comments and support, we really appreciate them. Here we address some of your concerns.
>
> **Q1. Where is Definition 5, and what does it mean by proof of Definition 5?**
>
> A1. Sorry for the confusion, Definition 5 means Definition 4-1. We will fix it in the revised version, and separate it into definition and proposition. Right now Definition 4-1 defines what $\alpha$ policy is, and suggests some mathematical properties of it (which should be proposition), which we prove in the "Proof of Definition 5". We will clearly mention that the proof in line 419 is for the proposition part of Definition 4-1.
>
> **Q2. Could you please explain / clarify the fluctuation of the blue curve (i.e., GI-PPO) in Figure 3 (c)?**
>
> A2. Thank you for the question, the 1-d Ackley’s function that is dealt with in Figure 3 (c) has its optimum at $x = 0$. Therefore, the best policy would give us a mean of 0 and variance of 0 for this problem, as shown in Figure 1 (a). In Figure 1 (a), we can observe that the optimum is formed at mean = 0, var = 0, **but the optimum is very unstable** - that is, small perturbations in either mean or variance lead to a large degradation in the results. The fluctuation in the blue curve in Figure 3 (c) stems from this nature of the target function. However, since we use Adam optimizer, which adjusts its step size based on the optimization trajectory, we can observe that the fluctuation disappears in the end.
>
> **Q3. The order number (or citation) of Figures in this manuscript is messy.**
>
> A3. Thank you for pointing this out.   Due to the page limit, the figures are not always ordered and appear, as desired.  We will rearrange the figures in their placement within the latex file to better align figures with the page break in the manuscript, as suggested.
>
> **Q4. If possible, some theoretical results, for example, Proposition 4.5, can be verified by empirical experiments?**
>
> A4. Thank you for the suggestion, we can definitely verify it with additional experiments. We will incorporate those results in the final version.
>
> **Q5. There are miscellaneous typos / issues in the writing.**
>
> A5. Thank you for the detailed feedback!  We will further polish our exposition to incorporate your suggestions -- very much appreciated!

---

> > ### Comment · Reviewer_WNVi · 2023-08-12
> > **Thank the author(s) for the responses**
> >
> > Thank the author(s) for the responses. I read the reviewers' comments and the rebuttals from the author(s).
> >
> > The author(s) answered my questions and cleared up my doubts. I increase my score and vote to accept this manuscript, but on the condition that the author(s) will carefully make the changes and correct the issues in the final version, as claimed in responses/rebuttals. Thanks and good luck!

---

> > > ### Author Response · Authors · 2023-08-14
> > >
> > > Thank the reviewer, we really appreciate the support! We’ll carefully revise the paper, as the reviewers suggested. Thanks.

---

### Official Review · Reviewer_xDc8 · 2023-07-27

**Soundness:** 4 excellent
**Presentation:** 3 good
**Contribution:** 3 good
**Rating:** 7
**Confidence:** 4

**Summary:**

This paper introduces a policy gradient algorithm based on Proximal Policy Optimization (PPO) algorithm. This new algorithm utilizes the analytical gradients from differential environments and achieves competitive performance in various scenarios including function optimization, classic control environments, and traffic control environments.
To integrate the analytical gradient into the PPO framework, they use the reparameterization trick and introduce the concept of $\alpha$-policy which is a locally superior policy. When $\alpha$ is sufficiently small, the difference between the original policy and this $\alpha$-policy would be small enough to satisfy the difference constraint in the PPO algorithm. Moreover, they propose metrics considering variance, bias, and out-of-range-ratio to dynamically adjust $\alpha$ during the training.

**Strengths:**

1. The idea of utilizing analytical gradients in physical control environments is powerful and can dramatically increase the efficiency of policy gradient algorithms when analytical gradients are available. This paper advances this idea by integrating it with the PPO algorithm and achieves competitive performance in various environments.

2. This paper proposes the concept of $\alpha$-policy and connect it with the reparameterization trick. $\alpha$-policy is an adjustable policy such that when $\alpha$ is small enough the difference between $\alpha$-policy and the origin policy can be small enough to fit into PPO algorithm.

3. This paper proposed metrics considering variance, bias, and out-of-range-ratio to dynamically adjust $\alpha$ during training.

**Weaknesses:**

1. When trying to fit the analytical gradients into the PPO algorithm, this paper didn't consider dynamically changing the PPO clip limit. Thus, the analytical gradients are not fully utilized.

2. More theoretical analysis and better ways on how to adjust $\alpha$ during training could be done and proposed.

3. The experimental performance showed improved performance compared to PPO but does not achieve state-of-art performance in some environments.

**Questions:**

1. Line 419 "Proof of Definition 5" does not match the actual definition index. The author should separate definition 4.1 into a definition and a proposition.

2. Can the author integrate the variance reduction techniques in [Xu et al., 2022] to achieve better performance? If not, what changes can be made to accommodate this?

**Limitations:**

I did not identify any potential negative societal impact from this work.

---

> ### Author Rebuttal · Authors · 2023-08-08
>
> Thank you for your valuable comments and support, we really appreciate them. Here we address some of your concerns.
>
> **Q1. The analytical gradients are not fully utilized, because this paper didn't consider dynamically changing the PPO clip limit.**
>
> A1. Please refer to A1 of global rebuttal at the top.
>
> **Q2. More theoretical analysis and better ways on how to adjust $\alpha$ during training could be done and proposed.**
>
> A2. Please refer to A3 of global rebuttal at the top.
>
> **Q3. The experimental performance showed improved performance compared to PPO but does not achieve state-of-art performance in some environments.**
>
> A3. Please refer to A2 of global rebuttal at the top.
>
> **Q4. Line 419 "Proof of Definition 5" does not match the actual definition index. The author should separate definition 4.1 into a definition and a proposition.**
>
> A4. Sorry for the confusion, and thank you for pointing it out. Definition 5 means Definition 4-1. We will fix it in the revised version, and separate the definition into definition and proposition as you commented for better understanding.
>
> **Q5. Can the author integrate the variance reduction techniques in [Xu et al., 2022] to achieve better performance? If not, what changes can be made to accommodate this?**
>
> A5. Thank you for the suggestion! In fact, **we already applied techniques in [Xu et al., 2022] to the analytical gradients that we used for GI-PPO in this work**. To be specific, we used the truncated time window suggested in the paper to estimate the analytical gradients and the following $\alpha$ policy. We will add additioinal experimental results that show how the length of truncated time windows affect the training results in the supplementary material.

---

> > ### Comment · Reviewer_xDc8 · 2023-08-17
> > **Thank the authors for the responses and supplementary materials**
> >
> > I would like to thank the authors for the responses and supplementary materials. They have addressed my questions.
> >
> > I encourage the authors to integrate discussions in supplementary materials to the main paper because they provide more direct insight on $\alpha$.

---

> > > ### Author Response · Authors · 2023-08-17
> > >
> > > Thanks for the support, we will revise the paper as the reviewers' suggestions, thank you.

---

### Author Rebuttal · Authors · 2023-08-08

We really appreciate all of the valuable comments from reviewers!

Here we address the major concerns, with a 1-page supplementary document.  Our code will be released publicly, when the paper is published.

**Q1. This paper does not fully utilize the analytical gradients, because it does not consider dynamically changing PPO clip limit.**

A1.  This is a limitation of the current implementation. To dynamically update the PPO's influence, another metric is required to **compare PPO against analytical gradients**.  Even if we could evaluate the quality of analytical gradients in PPO framework, we need an entirely new perspective to do it in the opposite direction.   This new exploration is an exciting future research directions worthy of a full investigation on its own, as many other issues (pros & cons) needs to be considered.

**Q2. The experimental results showed that GI-PPO outperforms PPO, but does not achieve state-of-art performance in some environments, especially in physics environments.**

A2.  Our method clearly outperforms PPO in EVERY environment, but it could not achieve the best performance in some of them, especially in physics simulations (Ant, Hopper).  We'll briefly discuss reasons for this from several aspects.

(1)  We'd like to reiterate that the **RP method is proven to be much more effective at solving those physics problems than other methods, especially PPO**. This is because we have already applied the variance reduction techniques suggested in [1] to these problems. However, because of the reason described in A1, even if it is more advantageous to use analytical gradients over PPO, the performance of our algorithm might be constrained by PPO.

To validate this claim, we conducted additional experiments for the Ant environment, which are displayed in **Figure 1** of the attached document. In the figure, **it's evident that GI-PPO performs slightly better than the RP method when $\delta_{oorr} = 1.0$**. Observing how $\alpha$ and out-of-range-ratio evolve over time in the plots, we see that $\alpha$ is clearly constrained by $\delta_{oorr}$, and this places a detrimental effect on the learning process. If analytical gradients are indeed more beneficial, we can prioritize them over PPO. However, systematizing this approach is currently beyond the scope of our paper.  It is one of the key future research directions currently under consideration.

(2) It requires a more effective strategy to adjust alpha during training. **At present, our approach finds balance between computational cost and performance**.  (Please see A3 for details.)  We discovered that our algorithm's performance in the physics environment, especially in the Hopper environment, is hindered by this issue.  We aim to devise a more systematic strategy and will incorporate it, if feasible, in the paper's revision.

**Q3. There is not so much explanation for the algorithm design to adjust $\alpha$. There could be more theoretical analysis for the algorithm, and even better ways to adjust $\alpha$ during training.**

A3. There are indeed multiple strategies available to adjust the $\alpha$ parameter during the training process.

In the current paper, our chosen strategy involves adjusting $\alpha$ contingent on the optimization results of each iteration. This is achieved by multiplying or dividing it by a predetermined (hyperparameter) constant. The intent behind this is to employ the modified $\alpha$ in the subsequent iteration to better satisfy our proposed conditions. Therefore, there is no guarantee to meet the conditions in every iteration, though works well in practice. As an alternative, we experimented with fine-tuning the $\alpha$ parameter during every iteration to more closely adhere to the constraints.  However, our findings suggest that this approach entails a significantly higher computational expense relative to our primary approach -- thus with a tighter bound but less efficient in practice.

We recognize that superior methods likely exist beyond these techniques.  We continue to investigate possible improved methods grounded in the same principles, as suggested in this paper, with more theoretical analysis *and* better empirical results.  In the meantime, our proposed method here already shows considerably better performance than most of known baselines in diverse environments consistently and we also demonstrated the validity of our approach.

**Q4. There could be limitations in applying this method to practical use cases, because there is no analytical gradients in real world.**

A4. We’d like to introduce two key insights on how our algorithm is beneficial for such cases.  First, we’d like to note that our algorithm can leverage analytical gradients even when they are biased, as shown in SECTION 5.3, **while the original RP method cannot**.  This means that we can **roughly estimate the gradients even if they are not very accurate (e.g. using finite difference method) and leverage them for real world problems**.  Second, nowadays there are many approaches that try to distill policies learned in simulation with access to excess information, including analytical gradients, to student models with access to less information, which are deployed in real world [2]. More generally, policies trained robustly in simulation with analytical gradients can also be used as expert policies for other applications, such as model compression [3] or domain generalization [4].  Likewise, there are many benefits we can take advantages of using our approach, even when it comes to real world applications.

[1] Xu, Jie, et al. "Accelerated policy learning with parallel differentiable simulation."

[2] D. Chen et al., “Learning by Cheating”.

[3] A. Ashok et al., “N2N learning: Network to Network Compression via Policy Gradient Reinforcement Learning”.

[4] D. Li et al., “Learning to generalize: Meta-learning for domain generalization”.

---

### Comment · Area_Chair_Ezmh · 2023-08-19

To Authors:

Thank you for submitting your rebuttal. We appreciate your efforts in addressing the reviewers' comments and providing clarification.

To Reviewers:

We kindly request you to complete your response to the authors' rebuttal as soon as possible. Time is of the essence, and the deadline is fast approaching. Your timely feedback and expertise are crucial in ensuring a fair and thorough evaluation process. Please prioritize reviewing the authors' response and provide your final feedback accordingly.

---

### Decision · Program_Chairs · 2023-09-21

**Decision:**

Accept (poster)

**Comment:**

Weighting the pros and cons, overall, the reviewers believe this paper should be accepted.